# Brain-wide visual habituation networks in wild type and *fmr1* zebrafish

Emmanuel Marquez-Legorreta [1,11], Lena Constantin [1], Marielle Piber[2], Itia A. Favre-Bulle [1,3], Michael A. Taylor [4], Ann S. Blevins [5], Jean Giacomotto[1,6,7,8], Dani S. Bassett [5,9,10], Gilles C. Vanwalleghem [1,12✉] & Ethan K. Scott [1✉]

Habituation is a form of learning during which animals stop responding to repetitive stimuli, and deficits in habituation are characteristic of several psychiatric disorders. Due to technical challenges, the brain-wide networks mediating habituation are poorly understood. Here we report brain-wide calcium imaging during larval zebrafish habituation to repeated visual looming stimuli. We show that different functional categories of loom-sensitive neurons are located in characteristic locations throughout the brain, and that both the functional properties of their networks and the resulting behavior can be modulated by stimulus saliency and timing. Using graph theory, we identify a visual circuit that habituates minimally, a moderately habituating midbrain population proposed to mediate the sensorimotor transformation, and downstream circuit elements responsible for higher order representations and the delivery of behavior. Zebrafish larvae carrying a mutation in the *fmr1* gene have a systematic shift toward sustained premotor activity in this network, and show slower behavioral habituation.

[1] The Queensland Brain Institute, The University of Queensland, St Lucia, QLD 4072, Australia. [2] School of Medicine, Medical Sciences, and Nutrition, University of Aberdeen, Aberdeen AB25 2ZD, UK. [3] School of Mathematics and Physics, The University of Queensland, St Lucia, QLD 4072, Australia. [4] Australian Institute for Bioengineering and Nanotechnology, The University of Queensland, St Lucia, QLD 4072, Australia. [5] Department of Bioengineering, University of Pennsylvania, Philadelphia, PA 19104, USA. [6] Queensland Centre for Mental Health Research, West Moreton Hospital and Health Service, Wacol, QLD 4076, Australia. [7] Griffith Institute for Drug Discovery, School of Environment and Science, Griffith University, Brisbane, QLD 4111, Australia. [8] Discovery Biology, Griffith University, Brisbane, QLD 4111, Australia. [9] Departments of Electrical & Systems Engineering, Physics & Astronomy, Neurology, Psychiatry, University of Pennsylvania, Philadelphia, PA 19104, USA. [10] Santa Fe Institute, Santa Fe, NM 87501, USA. [11]Present address: Janelia Research Campus, Howard Hughes Medical Institute, Ashburn, VA, USA. [12]Present address: Danish Research Institute of Translational Neuroscience – DANDRITE, Nordic-EMBL Partnership for Molecular Medicine, Department of Molecular Biology and Genetics, Aarhus University, Aarhus, Denmark. ✉email: gilles.vanwalleghem@mbg.au.dk; ethan.scott@uq.edu.au

Habituation is a simple form of non-associative learning, characterized by a decrease in response after multiple presentations of a stimulus, that is conserved across much of the animal kingdom[1]. It allows animals to remain attentive to novel and ecologically relevant stimuli while minimizing their expenditure of energy on inputs that occur frequently without consequence. The strength and speed of habituation, and of recovery after periods without the stimulus, depend on the parameters of the stimulus and its repetitions (the intensity, frequency, and number of stimuli)[2,3]. Careful modulations of these stimulus properties have proven useful in exploring the relationships between repetitive stimuli and behavior, thereby providing clues about the underlying habituation circuitry[4–7].

Other work has addressed some of the molecular and cellular dynamics mediating habituation, including reductions in motor neurons' presynaptic vesicle release during short-term habituation and processes involving protein syntheses for longer-term forms of habituation[8–13]. Recently, studies exploiting high throughput methods have revealed a more complex molecular scenario, suggesting that multiple mechanisms contribute to the habituation process[14,15]. At the whole-brain level, fMRI studies in humans have revealed changes in activity for various brain regions during habituation[16–18]. The intervening scales, of regional circuits and inter-regional networks, cannot be addressed using targeted cellular techniques or traditional brain-wide approaches. These networks, and the ways in which they change during habituation, can only be addressed by observing activity in whole populations of neurons (up to and including the whole brain) at single-cell resolution.

In recent years, exactly this approach has become possible in zebrafish larvae through the use of genetically encoded calcium indicators and light sheet or 2-photon microscopy[19–24]. Since zebrafish larvae undergo behavioral habituation[25,26] they have been used for experiments with tactile, visual, and acoustic stimuli, exploring the genetic and molecular mechanisms of specific circuits[27–32]. Furthermore, they share important molecular underpinnings of habituation with other species[33–35]. All together, these features make them an appealing platform for exploring brain-wide habituation circuitry.

This approach requires a robust innate behavior that is subject to habituation. Looming visual stimuli, which simulate approaching predators, reliably elicit startle responses that are conserved from insects to humans[36,37], and repeated looms have been shown to produce habituation in various species[38–40]. When looming stimuli are presented to larval zebrafish, visual information converges in the tectum, where local circuits are proposed to calculate the imminence of a threat[40–42]. However, additional structures respond to looms[40–45], and others, including the hypothalamus, modulate the visual escape behavior in contexts other than habituation[43,46,47]. The result is an intriguing but rough outline of the habituation network, and in the absence of a whole-brain cellular-resolution analysis, numerous questions about this behaviorally important process remain unanswered. These include the functional categories of loom-responsive neurons located across the brain, their distributions across and within brain regions, and ways in which information passes through their networks before, during, and after habituation.

Addressing these questions is especially important because of the role that sensorimotor transformations and habituation play in psychiatric disorders including schizophrenia, autism spectrum disorder (ASD), and Fragile X syndrome (FXS)[48]. While these disorders are traditionally diagnosed around their social or cognitive symptoms, each has characteristic alterations in sensory processing, habituation, and sensorimotor gating that compound, or in some cases may drive, social and intellectual impairments[49,50]. FXS patients, for example, show slow habituation[51–54], a phenotype also found in *fmr1*-mutant mice that model FXS[55,56]. While fMRI and EEG studies have revealed some of the regional changes in neural activity that correlate with habituation deficits in various psychiatric disorders[57–60], the network-wide causes of these symptoms remain largely unexplored.

Here, we report brain-wide calcium imaging during visual learning in larval zebrafish as they habituate to repeated threatening loom stimuli. We show distinct populations of habituating neurons and their characteristic distributions across the brain. We then use graph theory to identify the network dynamics of these habituating populations of neurons and explore how a mutation in the *fmr1* gene affects these dynamics.

## Results

**Habituation of visual escape behavior in larval zebrafish**. To characterize the escape behavior of larval zebrafish exposed to looming stimuli, we designed a 12-well apparatus in which each well contained a larva receiving its own loom stimulus from below (Fig. 1a). We presented looms in blocks of 10, with five minutes between blocks and an auditory tone at the end of the second rest period (for dishabituation before the 21st loom stimulus). In order to explore the relationships between stimulus properties and behavioral habituation, we used looming stimuli of two expansion speeds (a fast stimulus that filled the bottom of the well in 2 s and a slow stimulus that took 4 s) and two interstimulus intervals (ISIs) of 20 or 60 s between looms. These parameter choices resulted in four stimulus trains: f20, f60, s20, and s60 (Fig. 1b).

Each led to habituation of loom-elicited startle responses (Fig. 1c and Supplementary Fig. 1), and two patterns arose across the four stimulus trains. First, the slow-growing stimuli led to stronger habituation than the fast stimuli did, especially in the first block of 10 looms. Second, the stimulus trains with 20 s ISIs produced faster habituation within blocks, but the stimulus trains with 60 s ISIs produced habituation that recovered less after the 5-min rest periods. A generalized linear mixed model (GLMM) of the first block indicated a significant effect of the loom presentation number ($\beta = -0.25365$, $p = 2.00 \times 10^{-16}$) on response probability, confirming habituation. The loom speed also affected response probability strongly ($\beta = -1.23839$, $p = 2.22 \times 10^{-8}$) with a weaker impact from the ISI ($\beta = 0.45089$, $p = 0.038$). Together, the speed, ISI, and presentation number explain almost 20% of the variance ($R^2 = 0.1864$), and together with the random variable (fish identity) the model explained more than 35% of the variance ($R^2 = 0.3647$). These effects are consistent with past studies in zebrafish and other diverse model systems[5,7,34,35], suggesting a relationship between stimuli and habituation behavior that is broadly conserved. Explaining this relationship requires an exploration of the underlying circuitry and the ways in which it changes during habituation.

**Brain-wide characterization of neural activity during habituation**. To address brain-wide patterns of activity during habituation and the types of individual neurons that drive them, we moved to a head-embedded preparation in which loom stimuli were presented on an LCD screen. We performed whole-brain imaging of the *elavl3:H2B-GCaMP6s* line using selective plane illumination microscopy (SPIM, see "Methods" section). For each larva, this produced 50 horizontal planes spanning the rostrocaudal and medio-lateral extents of the brain, at 5 μm intervals along the dorso-ventral axis, with a volumetric acquisition rate of 2 Hz. We performed segmentation of these images to identify regions of interest (ROIs) generally corresponding to individual

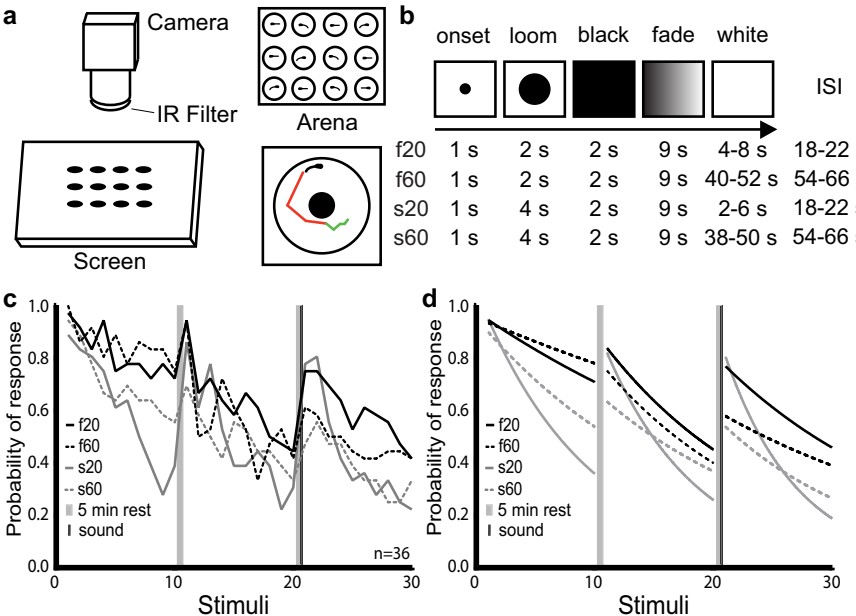

**Fig. 1 Modulation of habituation by stimulus features. a** Schematic representation of our setup for measuring visual habituation behavior. A 12-well chamber with one larva in each well (top right) was filmed on a horizontal screen (left) on which the looms were presented. Automated tracking recorded periods of swim bouts (green) and burst swim (red) for each larva (bottom right). **b** Stimulus train properties across the 4 experimental groups. In all cases, the stimulus appeared for 1 s before expanding over a 2-s (fast) or 4-s (slow) period. The resulting black screen was maintained for 2 s before fading to white over a 9 s period, followed by a variable period of white screen prior to the next stimulus period. The average ISI for each type of stimulus is shown in the right column: 20 s for f20 and s20, and 60 s for f60 and s60. ISIs were varied slightly to prevent the timed prediction of consecutive stimuli. **c** Probability of response across the 4 groups during three blocks of ten loom presentations. Probability was calculated at each loom presentation as: number of responding fish divided by total number of fish (n = 36). **d** Fitted exponential one-phase decay curves of the response probability for each group. The consistency of these results across different clutches of larvae is presented in Supplementary Fig. 1.

neurons, and extracted fluorescent traces from these ROIs (see "Methods" section and Supplementary Fig. 2).

Snapshots of responses across the brain during this repetitive stimulation (shown for f20 in Fig. 2a–c) show a sharp decrease in responsive ROIs between the first and second stimuli, and a further drop in responses by the 10th stimulus. Figure 2d, e shows the response of each ROI in the second and 10th trial as a proportion of its response in the first. Habituation is conspicuous across all loom-responsive brain regions, including the tectum, thalamus, medial hindbrain, tegmentum, and telencephalon, suggesting that these regions are affected by or involved in the habituation process.

To address the possible underlying mechanisms, we used k-means clustering to identify various categories of responsive neurons. We initially generated 50 categories (clusters) per data set, and then selected six clusters of loom-responsive neurons that were (1) not the products of imaging artifacts, (2) responsive to the first presentation of the loom stimulus, and (3) represented in a majority of the fish in the data set (see "Methods" section for numbers in each category). We also identified one auditory-responsive cluster, in which ROIs responded to a sound presented before the 3rd block of looms, but did not pursue these responses further as they failed to produce significant dishabituation. Based on our data, it is not possible to judge whether this particular acoustic stimulus is not appropriate for driving dishabituation, or whether zebrafish do not undergo such dishabituation. The simplest explanation is that our acoustic stimulus, which was delivered through an air interface, was not salient enough to cause dishabituation in this context, and that a stronger stimulus delivered directly to the water in the imaging chamber could have produced such an effect[61,62].

Based on their highly similar response properties, we merged three clusters of ROIs showing strong and rapid habituation (Supplementary Fig. 3) into a single strongly habituating cluster (Fig. 2f, g). We characterized the remaining three clusters as moderately habituating, weakly habituating, and inhibited, and we also located a motor-associated group of ROIs using regressors customized to each animal's movements (Fig. 2f, g). A t-SNE analysis (Supplementary Fig. 4) shows segregation among these clusters, supporting the idea that our clustering found distinct categories of loom-responsive neurons.

Strongly habituating ROIs are spread across several brain regions (Fig. 2h and Supplementary Movie 1, Supplementary Fig. 5), most prominently in the tectum, thalamus, medial hindbrain, pallium, and tegmentum. In the hindbrain, these ROIs are concentrated in a longitudinal rostro-caudal strip along the pathway of the tectobulbar projections, meaning that they likely include reticulospinal premotor neurons[63,64].

Moderately habituating ROIs are tightly concentrated in the central region of the tectal periventricular layer (PVL) of the left tectum (Fig. 2h and Supplementary Movie 1, Supplementary Fig. 5). This laterality is unsurprising, since the stimulus was presented to the right eye, and since all retinal projections are contralateral in zebrafish larvae. This position is consistent with a role for the associated neurons in the spatially registered processing of visual information, and their decreased responses may represent an important element of the overall circuit's reduced responsiveness during habituation.

Weakly habituating ROIs are prominent in the tectum, habenulae, pretectum, and pallium (Fig. 2h and Supplementary Movie 1, Supplementary Fig. 5). There is moderate laterality toward the contralateral side to the stimulus in most of these regions. In the pallium, responses are concentrated around the

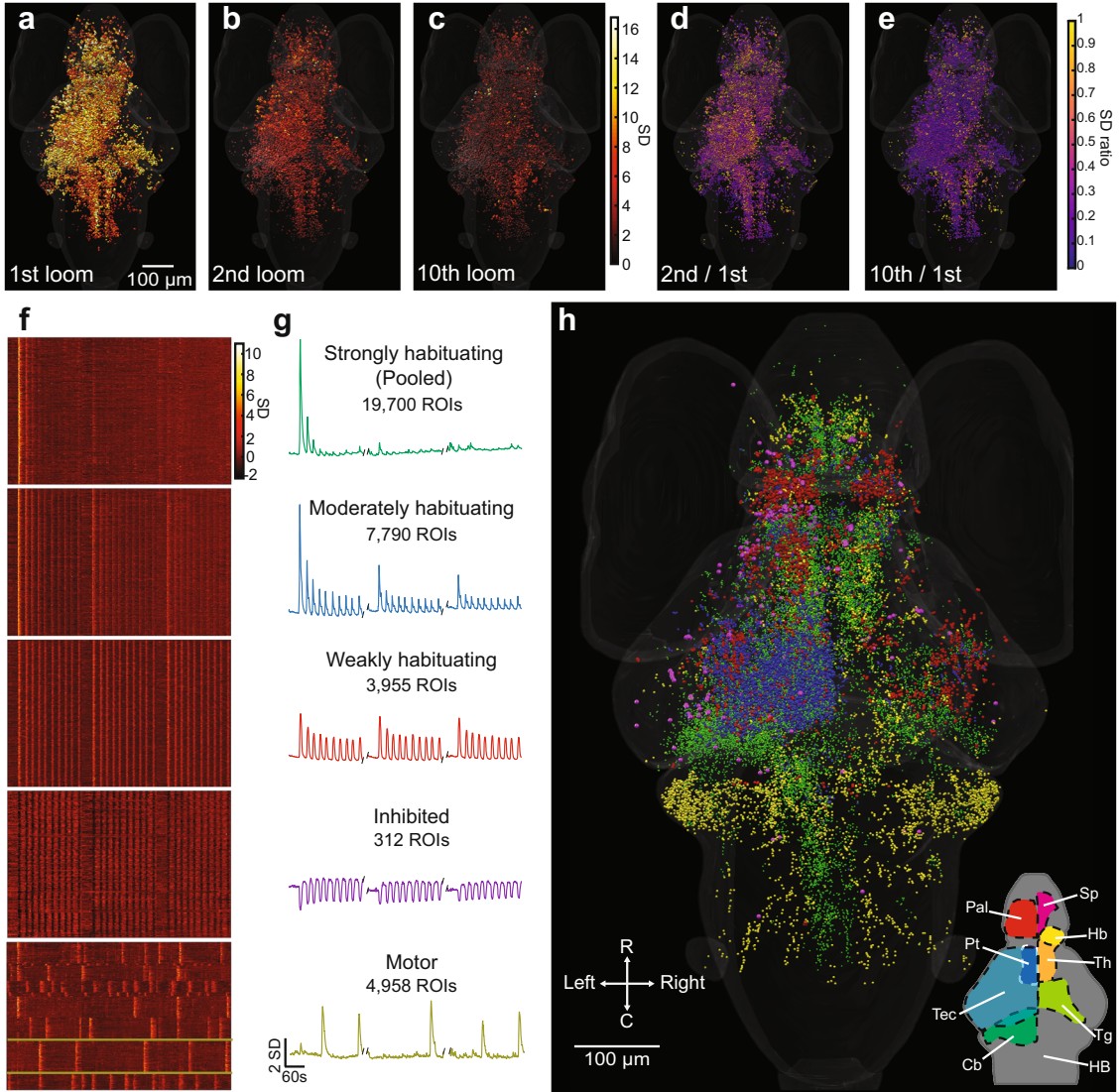

**Fig. 2 Activity of individual ROIs and their functional clusters during habituation. a** Responses of ROIs across the brain to a loom stimulus, color-coded for the normalized intensity of their response. **b, c** The same ROIs' responses to the second and tenth looms. **d, e** The degree of habituation in each of these ROIs in the second and 10th trials, calculated as the ratio of response to the first loom. This analysis was restricted to ROIs showing clear responses (with a coefficient of determination ($r^2$ value) > 0.5 for the linear regression between their response and a regressor simulating a calcium signal) for the first loom stimulus. Raster plots (**f**) and mean response traces (**g**) of the ROIs composing each of five functional clusters, with a clear correspondence to the three blocks of ten stimuli. The $x$ axis scale at the bottom of **g** also applies to **f**. **h** Anatomical locations for the ROIs belonging to each functional cluster (colors matching the mean traces in **g**). Since different animals startled in different trials, we identified the motor cluster using a different regressor for each animal. The mean responses are shown for a single animal in this cluster in **g**, with yellow lines indicating the relevant neurons from that animal in **f**. A rotation of **h** can be found in Supplementary Movie 1, and the distributions of these clusters are detailed in virtual sections in Supplementary Fig. 5. Data shows the pooled responses of 11 larvae to the f20 stimulus train. Relevant anatomical brain regions are indicated in the bottom right corner of **h**, each shown for only one side of the brain. Pallium, Pal; subpallium, Sp; thalamus, Th; habenula, Hb; pretectum, Pt; tectum, Tec; tegmentum, Tg; cerebellum, Cb; and hindbrain, HB. R, rostral; C, caudal.

dorsal edge of the pallium in what will likely become the lateral division of the dorsal pallium (Dl)[65,66], although they also extend into the medial division (Dm, Supplementary Movie 1 and Supplementary Fig. 5).

Inhibited ROIs are rare and mostly localized to the contralateral tectum and rostral thalamus (Fig. 2h and Supplementary Movie 1, Supplementary Fig. 5). Motor-associated ROIs are concentrated in the cerebellum. However, some can be found in the anterior and lateral hindbrain and small numbers occur in the thalamus and pallium (Fig. 2h and Supplementary Movie 1, Supplementary Fig. 5). These ROIs are presumably involved in the coordination and delivery of the escape responses. It is

important to note that we did not observe consistent escape responses during our brain imaging experiments, this is likely due to the immobilized fishes' switching to a passive state when their behavior does not trigger any perceived change of position or relief from the looming stimulus[19].

**Temporal stimulus properties influence ROIs' responses, but not their distributions**. The fundamental brain-wide habituation network was conserved across our four habituation paradigms (Fig. 3), with a few specific differences. One was a greater number of strongly habituating ROIs in the hindbrain for the s20 and s60 experiments. Another came in experiments with 60 s ISIs, where

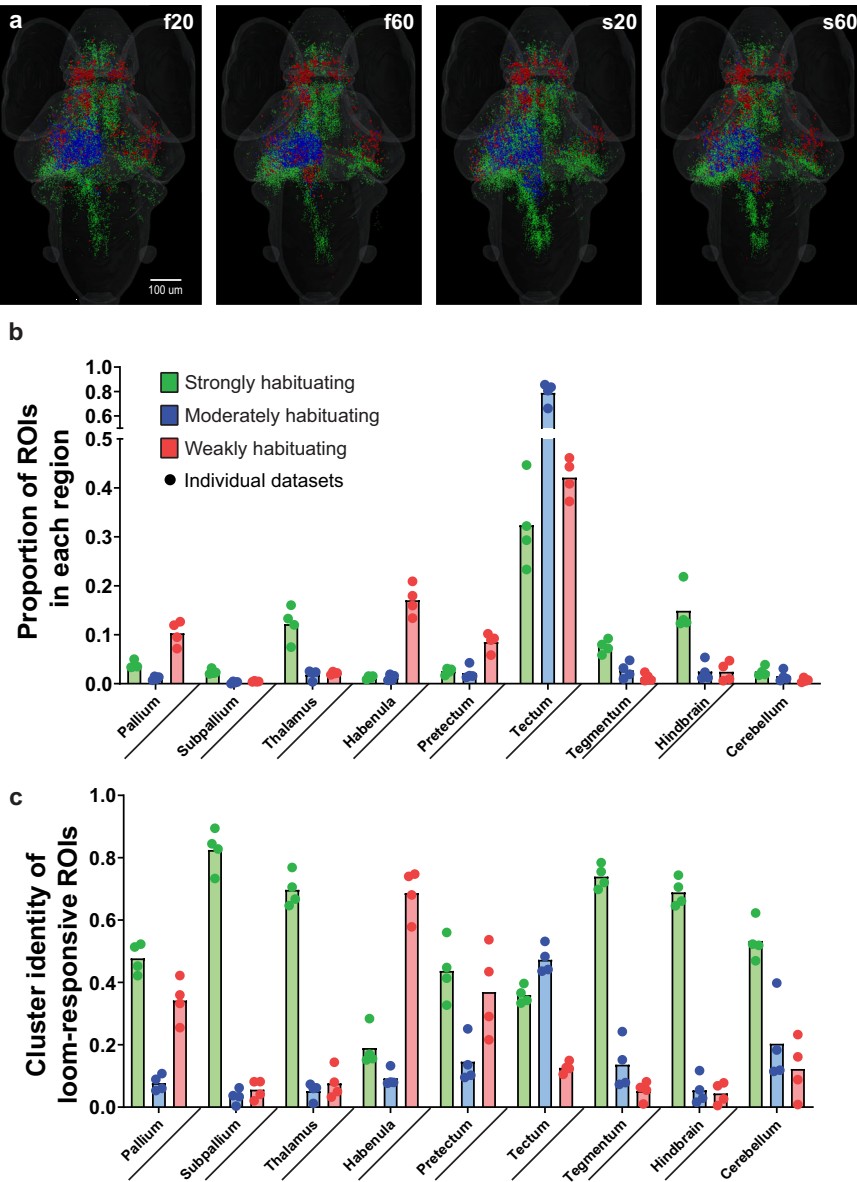

**Fig. 3 Brain-wide responses during different loom stimulus trains. a** Brain-wide distributions of the functional clusters from Fig. 2 for each of four loom habituation paradigms. **b** Averaged proportions of ROIs in the four data sets, from the brain's total number of each functional cluster, located in the indicated brain regions. **c** Averaged proportion of each cluster's representation among the loom-responsive ROIs in each brain region. The individual paradigms (f20, f60, s20, and s60) are represented as circles, $n = 4$ paradigms. Statistical analysis of these proportions is detailed in Supplementary Tables 1 and 2.

we observed a greater number of weakly habituating ROIs in the dorsal hindbrain on the side contralateral to the stimulus (Fig. 3a). This hindbrain neuronal population's differences most likely reflect changes in neuronal activity linked to the stimulus features which ultimately influence to which functional cluster each ROIs is included. However, the proportion of ROIs of each cluster distributed across brain regions is very similar and is consistent with the results described above for f20, which suggests a distinct distribution of the habituating population subtypes (Fig. 3b). Similarly, within each brain region, there is a characteristic abundance of each cluster that is essentially preserved across all four paradigms (Fig. 3c).

The facts that the habituating clusters have distinct distributions, and that these distributions are preserved in different paradigms, suggests that each cluster and region plays a distinct role in the habituation circuit (although these roles can be

modulated somewhat by the details of the stimuli, and some localized populations may represent a continuum of response properties). If our methods were detecting various points on a continuum of responses, we would expect to see more diffuse localizations of the habituating clusters across data sets. The constant responses of the weakly habituating ROIs suggest that these neurons are part of a core visual pathway sensitive to the features of the loom (luminance change and/or moving edges). The tight localization of moderately habituating neurons in the tectum, and the facts that they habituate gradually and are more sensitive to the recovery (Fig. 2g), suggest that these neurons are integrating the current and past visual information, possibly reflecting a significant element in the overall circuit. Finally, the strongly habituating ROIs present a wider distribution across the brain and seem to include not only sensory processing regions but also premotor and upstream processing areas of the brain. This

suggests that the corresponding neurons are part of a threat detection response pathway and cease to respond quickly as the stimulus is no longer perceived as a threat. This idea is reinforced by the fact that the strongly habituating ROIs overlap with regions that are also sensitive to auditory stimuli (Supplementary Fig. 6a, b).

By considering the changes in the activity across our four habituation paradigms (Fig. 1), we next aimed to identify the patterns of brain-wide activity which best represented the different rates and persistence of habituation in free-swimming larvae. A correlation between the activity of each brain region with the free-swimming responses of its matching stimulus group shows that moderately habituating neurons are most likely to resemble the responses presented in the behavioral experiments and that the tectum presents the highest correlations for each functional cluster (Supplementary Fig. 6c). We next considered the relationship between the stimulus train's properties and the responses of each functional cluster in this structure (Supplementary Fig. 6d). This analysis revealed only subtle differences across the stimulus trains for the response profiles of fast habituating neurons. For moderately habituating neurons, differences arose with intriguing parallels to the free-swimming behavioral outputs. In experiments with 60 s ISIs, habitation was slower, and recovery was less dramatic than for 20 s ISI experiments. In experiments that used slow stimuli, habituation occurred faster than in the corresponding experiments with fast loom stimuli. We tested these observations using a nonlinear regression to fit curves to the normalized responses during the first block of habituation (Supplementary Fig. 6e). The results show that moderately habituating neurons within the tectum are significantly different for the rate constant and plateau values across data sets ($F(6388) = 9.778$; $p < 0.0001$). In experiments with 60 s ISIs, habituation is slower (represented by a higher Plateau and lower rate constant [Plateau: f20 = 0.1889; f60 = 0.2305; s20 = 0.1728; s60 = 0.2240. rate constant K: f20 = 0.9916; f60 = 0.8387; s20 = 1.019; s60 = 0.99]), and recovery is less dramatic than for 20 s ISI experiments (Supplementary Fig. 6f, 10th vs 11th looms predicted mean diff. and adjusted $p$ values of Sidak's multiple comparisons test: f20 = −0.2339, $p = <0.0001$; f60 = −0.04707, $p = 0.5405$; s20 = −0.2063, $p \leq 0.0001$; s60 = −0.1118, $p = 0.0019$). For weakly habituating neurons, experiments with 60 s ISIs led to less habituation throughout the experiment, while other correlates of free-swimming behavior were less clear. Overall, moderately habituating ROIs repeatedly had the strongest correlation to free-swimming escape probability (Supplementary Fig. 6g, mean Pearson correlation values: f20 = 0.672, 95% CI [0.6421, 0.702]; f60 = 0.6744, 95% CI [0.6269, 0.722]; s20 = 0.644, 95% CI [0.5898, 0.6983]; s60 = 0.6443, 95% CI [0.613, 0.6757]), suggesting that among our functional clusters, it is the moderately habituating ROIs in the tectum whose dynamics most closely reflect behavior.

**Network modeling of visual loom habituation**. As an approach for modeling visual loom processing and the network changes that produce habituation, we used methods from graph theory, which are well suited to analyzing brain-wide activity data[67–72]. To allow comparisons across fish and groups, we created common reference points that preserved the anatomical location and functional identity of the loom responses. We spatially clustered the 144,709 responsive ROIs into 99 nodes that represent the ROIs' functional clusters and their associated anatomical locations, and then we produced matrices representing the correlations in activity between all pairs of these nodes at different times during the experiments (Fig. 4a, see "Methods" section). Each correlation matrix encodes a graph composed of functional

relations (edges) between pairs of grouped ROIs (nodes). The validity of this method was confirmed by demonstrating robustness to the number of nodes (Supplementary Fig. 7), by comparison to null models (Supplementary Fig. 8), and by leave-one-out cross-validation (Supplementary Fig. 8). We then compared these matrices in larvae exposed to the f20 and f60 habituation paradigms to identify the graph-level correlates of behavioral habituation (Fig. 4). As expected, both paradigms produced high correlation values in response to the first loom, and the matrices for the two paradigms were highly similar. As habituation proceeded, graph correlations remained somewhat higher in the f60 paradigm, reflecting differences in the behavioral responses during the f20 and f60 experiments (Fig. 1c, d). By the 10th loom, most of these correlations had dropped dramatically for both paradigms, with high values mostly restricted to correlations between weakly habituating (red) nodes. The f20 paradigm shows a stronger recovery across the graph in the 11th trial, reflecting the stronger behavioral recovery that takes place in this paradigm.

As an approach to judge both the rate at which these correlations were lost during the first block of stimuli and the degree to which they recovered in the 11th trial, we used a Pearson correlation to match the matrix of the 11th trial to the most closely related matrix from the first block of stimuli (Fig. 4a). The highest Pearson correlation coefficients were for the 4th trial for f20 and the 6th trial for f60, indicating both that the correlations are lost more quickly in f20 (the paradigm in which habituation occurs more quickly), and that the recovery is weaker in f60 (the paradigm that produces more indelible behavioral habituation). Notably, the patterns of correlations across the matrices during mid-habituation trials (4th for f20, Fig. 4a) strikingly resemble those in the 11th trials, suggesting that the network is returning to a partially habituated state, rather than assuming a distinct post-habituation state.

These results show that the loss of correlations across nodes in the graph reflects free-swimming behavioral habituation dynamics. To describe the graphs, we represented them spatially and mapped the relative correlation strengths between nodes in the f20 and f60 paradigms (Fig. 4b). Each edge (node-to-node relationship) in the graph is represented by its correlation value in the f20 paradigm minus its value in the f60 paradigm. As expected, because the first trial is identical, both paradigms show robust correlations across numerous edges in the first trial, with most edges near a zero value and no net weighting of the graph toward positive or negative. By the 10th trial, the graph has lost most edges, and the remaining activity is biased toward stronger correlations in f60 (shown in red), reflecting the slower habituation. The f20 paradigm shows the stronger recovery, however, and this effect is captured in a shift toward positive values (blue) in the 11th trial.

Changes in the correlations between different functional categories of neuron are of particular interest, as they could indicate which specific correlations contribute to loom responses, and by association, to habituation. Therefore, our next goal was to quantify the level of functional connectivity between different clusters, and how this connectivity changes after repeated loom presentations. We quantified the participation coefficient of each node in the graph, which is defined as the proportion of a node's edges that are shared with nodes from a different functional cluster (as defined in Fig. 2). The participation coefficient dropped over the course of 10 stimuli (Fig. 4c), but this drop was slower in f60 than f20, suggesting that habituation is driven not only by a drop in correlated activity across nodes, but specifically by a loss of communication between different functional clusters. This conclusion is reinforced by the higher participation coefficient in the 11th trial of the f20 paradigm, where strong behavioral recovery is echoed by a recovery in

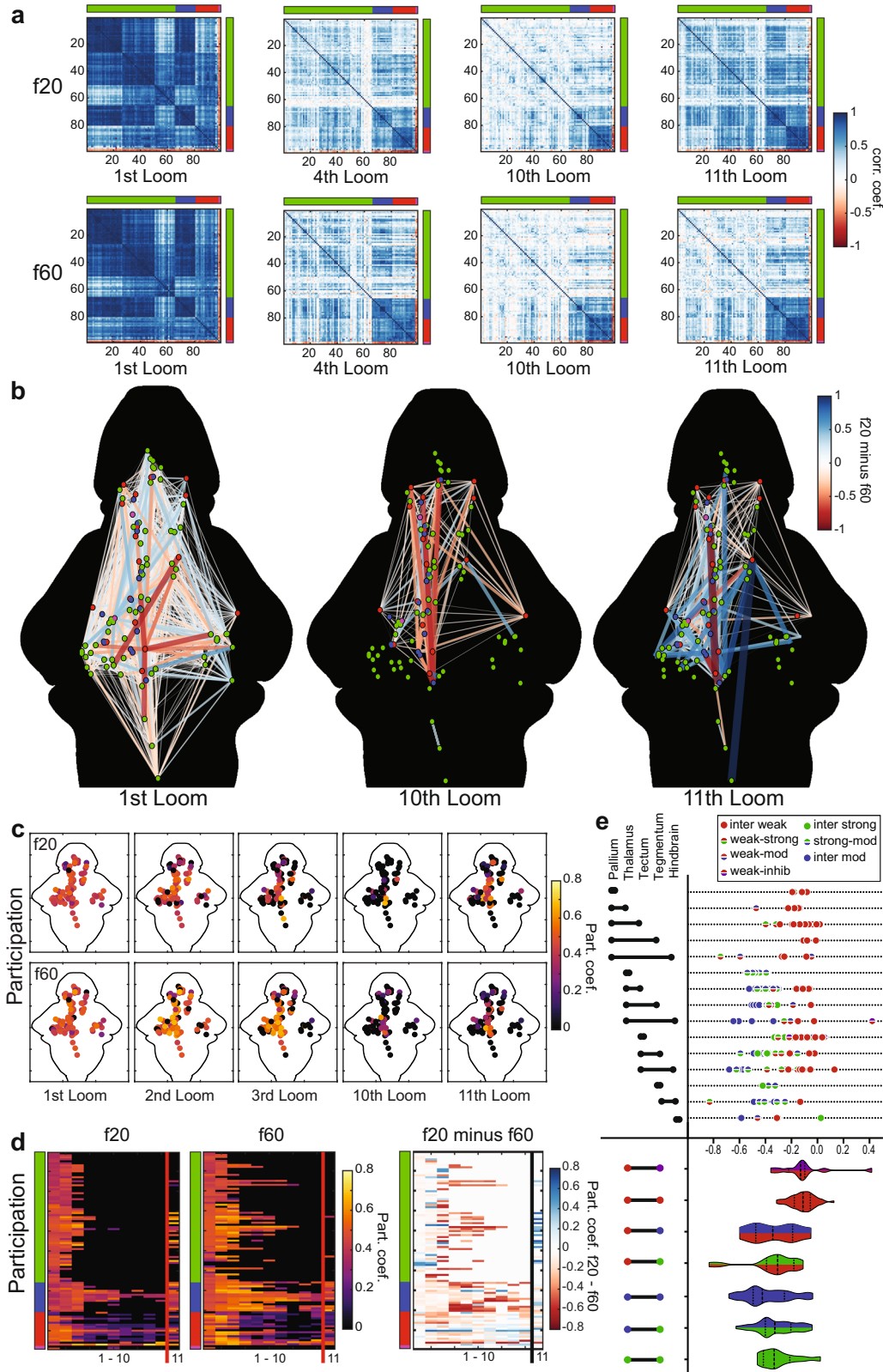

participation coefficient. Raster plots of the participation coefficient by each node across the first 11 trials (Fig. 4d) show this trend, further suggesting that it is weakly habituating (red nodes) that maintain much of their participation across functional clusters as habituation proceeds, and that recovery is accompanied by a resumption of such participation by various strongly (green) and moderately (blue) habituating nodes.

To address which brain regions are involved in this process, we mapped the edge strength (degree of correlated activity between nodes) across five regions containing a majority of the nodes (the pallium, thalamus, tectum, tegmentum, and hindbrain; Fig. 4e). The values for each edge, represented by a dot, show the correlation in the 10th trial minus the correlation in the 11th trial, thus giving negative values to edges that became stronger during

**Fig. 4 Graph model of the visual loom network during habituation. a** Correlation matrices for activity of 99 nodes representing ROIs across the whole brain. The functional clusters to which each node belongs are indicated on the axes, using the color code from Fig. 2. Darker blue shades represent stronger positive correlations for any given node pair, and red indicates negative correlations (see color scale, **a**). **b** A graphic representation of correlations across the 99 nodes, whose functional clusters are indicated by their colors and anatomical positions represented spatially. The colors and width of the lines indicate the relative correlation across the f20 and f60 experiments (f20 correlation minus f60 correlation), where red indicates stronger correlations in f60 and blue indicates stronger correlations in f20 (see color scale). Only edges with correlations above 0.75 in either the f20 or the f60 matrices are displayed. **c** A heat map of the participation coefficient for each of the 99 nodes during the 1st, 2nd, 3rd, 10th, and 11th loom stimuli of the f20 and f60 experiments. **d** Raster plots showing the participation coefficient of each node across the first 11 stimuli for f20 and f60, and the relative participation (f20 value minus f60 value) where blue indicates stronger f20 participation and red indicates stronger f60 participation. The functional clusters for each node are indicated, using the color code from Fig. 2. **e** Changes in correlation strength for edges from the 10th to the 11th looms of f20, indicating the impact of the recovery from habituation. Values shown are calculated for each edge as its strength in the 10th loom minus its strength in the 11th loom, with more negative values indicating edges that showed more pronounced recovery between the 10th and 11th looms (top). The functional clusters for each edge's two nodes are color-coded and the brain regions that the edges span are indicated on the left. Violin plots (bottom) show the cumulative distributions of edges connecting different types of functional clusters (left). Dashed lines indicate the median and dotted lines indicate the first and third quartiles.

recovery. Violin plots show the total distributions of edges between different functional clusters. The results confirm that certain types of edges, especially those between two weakly habituating (red) nodes, play a relatively small role in recovery, owing to their strong unhabituated responses in the 10th trial. Other types of edges, especially those not including a weakly habituating node, tend to have highly negative values, indicating that they contribute to the part of the graph that is lost during habituation and regained during recovery.

Collectively, these results converge toward a model of the brain-wide network that produces visual escape and the mechanisms by which these responses are suppressed during learning. The initial process of habituation appears to rest on the loss of correlation (and presumed communication) among neurons of different functional clusters. This segregation is manifested as a dramatic drop in correlation values for edges between different clusters (Fig. 4a), a restriction of the active graph principally to edges between nodes of the same type (especially weakly habituating nodes, Fig. 4b, e), and a loss in functional participation across clusters during the course of habituation (Fig. 4c). The striking similarity between mid-habituation matrices and those of partially recovered graphs (Fig. 4a) indicates that the matrix changes that underlie habituation are the same as those that are reversed during recovery, suggesting that the onset of habituation works through the same circuit-level changes that recovery does, and that there are not separable network-level mechanisms for the acquisition and retention of behavioral habituation.

**$fmr1^{-/-}$ mutant larvae show behavioral and network-level habituation deficits.** To test the validity and explore the utility of this proposed network, we next considered a zebrafish model of FXS, an inherited disorder characterized by altered habituation, intellectual disability, social deficits, and altered sensory processing. We used a nonsense mutation in the highly conserved $fmr1$ gene, the silencing of which causes FXS in humans. Given the learning deficits, including slow habituation, in humans with FXS[52–54], we explored whether and how behavior and brain-wide habituation networks are altered in $fmr1$-mutant zebrafish which presents developmental and behavioral phenotypes in line with mammalian models[73–77].

Using the s20 habituation paradigm in our free-swimming preparation, we found that $fmr1^{-/-}$, $fmr1^{-/+}$ heterozygotes (hets), and wild-type (WT) siblings share a similarly high probability of startling to the first loom stimulus (Fig. 5a). Habituation is slower, however, with a significantly higher response probability in the 2nd trial, and a trend towards greater response in the 3rd and 4th trials. There is also a stronger recovery after a break in $fmr1^{-/-}$ animals, although not significantly so.

Heterozygotes show an intermediate phenotype. The slowed habituation in $fmr1^{-/-}$ larvae is likely maladaptive in a natural environment, as escape responses to non-threatening stimuli waste energy and make the larvae conspicuous to other potential predators in the area. Furthermore, these behavioral results match the habituation deficits found in other animal models and human subjects with FXS. Such hyperresponsiveness could also be related to elevated levels of anxiety seen in subjects with FXS[78,79].

We next assessed correlates of this behavior using brain-wide calcium imaging, initially by considering the distributions of ROIs belonging to functional clusters (Fig. 5b). While all genotypes had fundamentally similar distributions, there was a trend toward more numerous weakly habituating ROIs in the cerebellum in $fmr1^{-/-}$ larvae, and toward a reduction in strongly habituating ROIs in the hindbrain, although neither of these trends was significant.

Turning to the graph representation of these results (Fig. 5c), we first examined correlations among 90 nodes (having eliminated nine of the original 99 nodes with a requirement that all nodes be represented in at least three larvae). Generally, correlations across the graph were stronger in WT than in $fmr1^{-/-}$ in the first trial (resulting in positive values shown in blue). This trend is reversed in the 2nd and 3rd trials, where the WT graph habituates more quickly, leaving negative (red) values that indicate persistent $fmr1^{-/-}$ network activity (Figs. 5c, 6a). Consistent with behavioral data, the overall correlations across the WT and $fmr1^{-/-}$ graphs are similar by the 10th trial, but WT graphs are stronger across the core perceptual pathway (tectum, thalamus, and pallium) described above, while $fmr1^{-/-}$ correlations are stronger across edges that habituate quickly in WT. Again echoing a trend in the free-swimming behavior, the $fmr1^{-/-}$ animals show dramatically broader and stronger pairwise correlations between nodes in the 11th trial, following a break in the stimulus. All of these observed differences carry through to measurements of participation across the different loom graphs (Fig. 5d, e).

By assessing correlation strengths across the graph in a way that represents nodes' functional and anatomical properties, we then outlined the overall functional architecture of the habituating $fmr1^{-/-}$ brain versus WT. First, we organized our brain-wide node-to-node relationships by functional cluster (Fig. 6b, c), allowing the level of correlation within and across clusters to be assessed. This structuring of the data shows that by the 2nd stimulus, there are still strong functional connections among red-red edges and along blue-blue edges in WT, and that these connections are largely restricted to red-red edges by the 3rd trial. By the 10th trial, strong correlations only exist in red-red edges (and a few to inhibited nodes, shown in purple). A subset of red-blue, blue-blue, and blue-green nodes reconnect in the 11th trial,

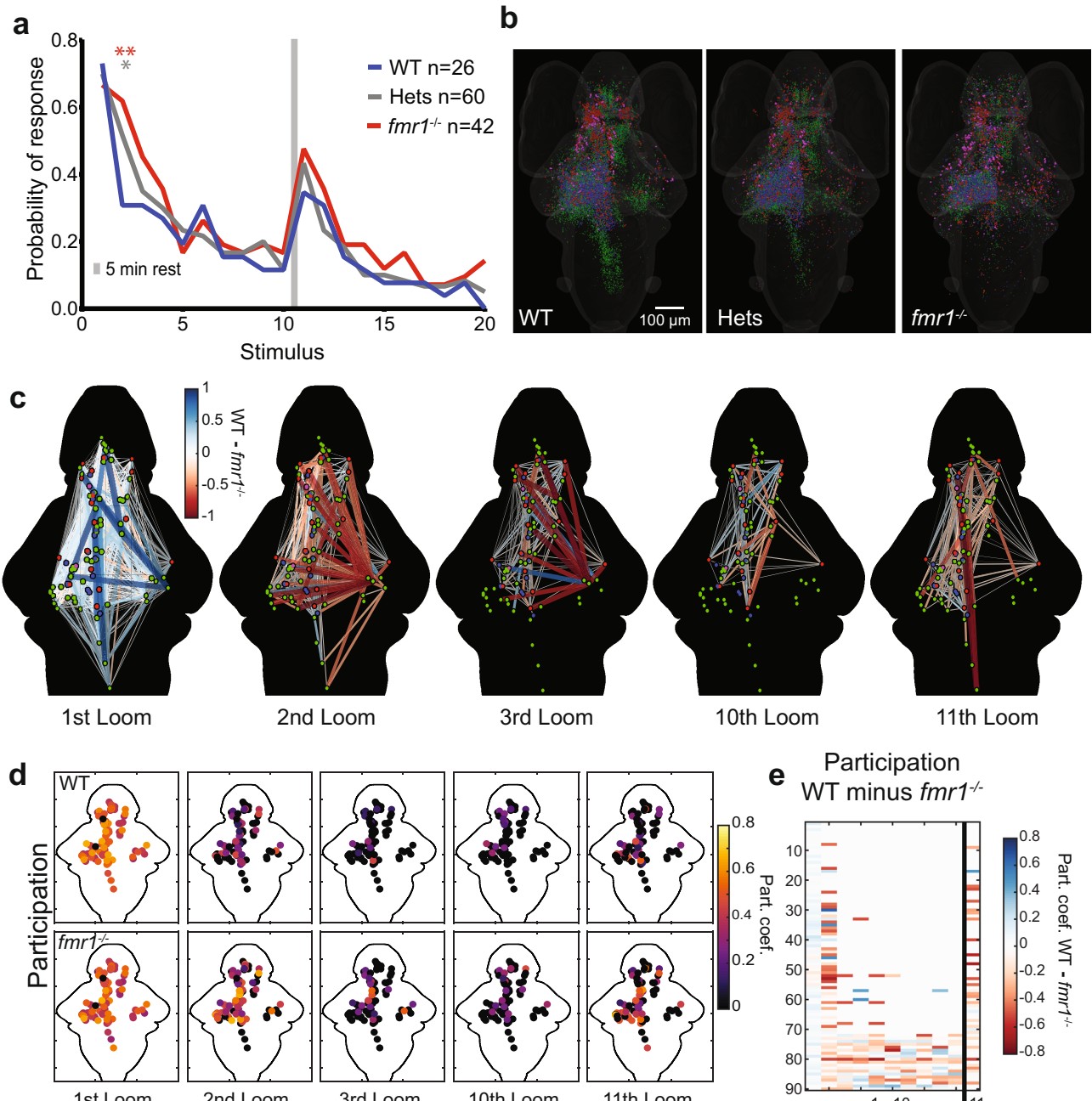

**Fig. 5 Behavioral and network-wide changes in *fmr1*−/− larvae. a** Probability of response across the three groups to two blocks of ten looms. Over the course of two blocks of 10 stimuli, *fmr1*−/− larvae show slower habituation and stronger recovery than WT siblings, and heterozygotes show an intermediate phenotype. One-sided binomial test: *fmr1*−/− versus WT: 2nd Loom ($p = 3.056e^{-5}$); 3rd Loom ($p = 0.034$); 11th Loom ($p = 0.055$); 16th Loom ($p = 0.039$). Heterozygotes versus WT: 2nd Loom ($p = 0.001$) and 9th Loom ($p = 0.039$). All other comparisons ($p > 0.1$). Significance cutoffs using a Bonferroni correction are *$p < 0.00125$ and **$p < 0.00025$, equivalent to uncorrected $p < 0.05$ and 0.01, respectively. **b** Brain-wide distributions of ROIs for the three genotypes, color-coded for functional cluster as in Figs. 2 and 3a. A random sample ($n = 11$) of Hets was selected to match WT ($n = 10$) and *fmr1*−/− ($n = 11$). **c** Node-based graphs showing relative correlations (WT correlation minus *fmr1*−/− correlation), where blue indicates correlations that are stronger in WT and red indicates correlations that are stronger in *fmr1*−/−. Larvae imaged to generate the graphs for WT = 10 and *fmr1*−/− = 11. **d** Heat maps of participation for all nodes across habituation and recovery. **e** A raster plot of relative participation (WT participation minus *fmr1*−/− participation) for each node through the first 11 trials.

reflecting recovery. In all regards, these effects resemble the habituating graph dynamics shown for the f20 paradigm in Fig. 4, where habituation tracks with a loss of communication between weakly habituating (red) nodes and strongly habituating (green) nodes, connected through moderately habituating (blue) nodes. By comparison, *fmr1*−/− animals show more strong correlations, and therefore more edges, between numerous nodes in the 2nd

and 3rd trials, as well as following recovery in the 11th trial (Fig. 6b, c). The distribution of the correlated edges is similar between the genotypes in the first and 10th trials (Fig. 6b), showing that the graphs are similar in the naive state and following habituation. Consistent with the analyses in Figs. 5c and 6a, this finding suggests that uncoupling across functional clusters occurs more slowly and recovers more completely in

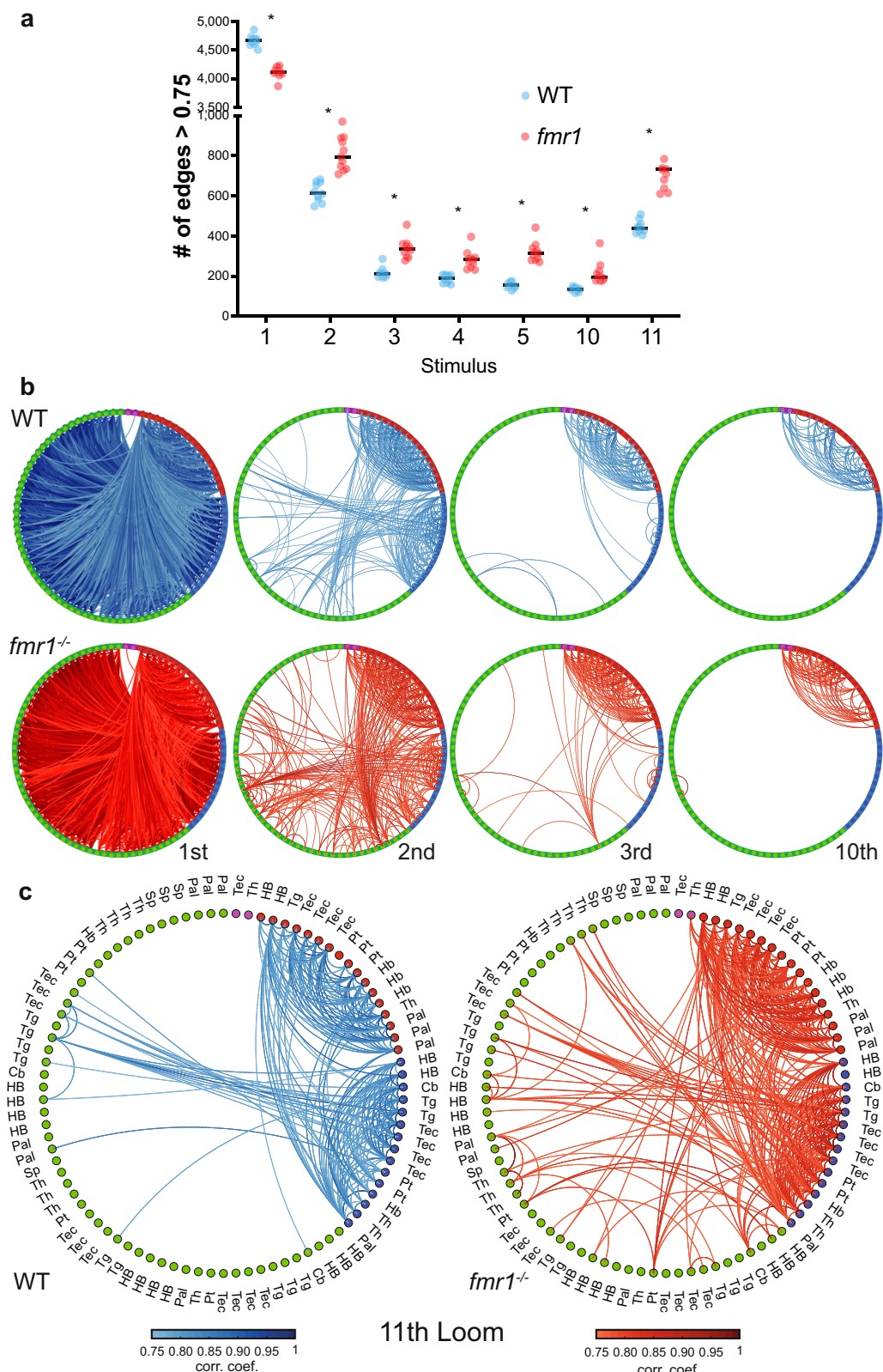

*fmr1*[−/−] animals, providing a mechanism by which the sensorimotor transformation is slanted toward downstream network activity and behavioral responsiveness in these animals.

To explore the spatial properties of this phenotype, we next represented these data organized by brain region (Supplementary Fig. 9). In WT animals, this structuring of the data makes clear that extensive correlation between nodes across all brain regions exists in the first trial (Supplementary Fig. 9), and as habituation proceeds, the correlation progressively winnows to the core perceptual circuit described above: mainly connections among the tectum, thalamus, and pallium on the side contralateral to the stimulus. In the 2nd, 3rd, and 11th trials (and to a lesser degree, the 10th trial), this network contains more connections in *fmr1*[−/−] animals (Fig. 6a and Supplementary Fig. 9), showing

**Fig. 6 Graph structure of WT and _fmr1_⁻/⁻ habituation networks. a** Comparison of the average number of edges >0.75 at different loom presentations using the leave-one-out approach to generate group-averaged matrices for WT (_n_ = 10) and _fmr1_⁻/⁻ (_n_ = 11). WT presents more edges at the 1st loom and _fmr1_⁻/⁻ fish at the following loom presentations, including the first loom after recovery (the 11th). Repeated measures two-way ANOVA (one-sided) with the Geisser–Greenhouse correction followed by a Šidák's multiple comparisons test. WT vs _fmr1_ > 0.75 edges Predicted mean diff., _p_ values of multiple comparisons test (and Šidák's correction adjusted _p_ values): 1st Loom = 546.9, _p_ ≤ 0.0001 (_p_ ≤ 0.0001); 2nd Loom = −192.3, _p_ ≤ 0.0001 (_p_ ≤ 0.0001); 3rd Loom = −121.7, _p_ ≤ 0.0001 (_p_ ≤ 0.0001); 4th Loom = −94.98, _p_ ≤ 0.0001 (_p_ = 0.0001); 5th Loom = −165.6, _p_ ≤ 0.0001 (_p_ ≤ 0.0001); 10th Loom = −81.07, _p_ = 0.0006 (_p_ = 0.0039); 11th Loom = −311.1, _p_ ≤ <0.0001 (_p_ ≤ 0.0002). Black horizontal bars indicate the median. **b**, **c** Functionally sorted brain-wide graphs for WT and _fmr1_⁻/⁻ larvae. Edges with correlations above 0.75 are shown between all combinations of nodes, and nodes are arranged by their functional clusters (Green: Strongly habituating; Blue: Moderately habituating; Red: Weakly habituating; Magenta: Inhibited). Graphs are shown for trials 1, 2, 3, and 10 (**b**), and trial 11 (**c**). The brain region to which each node belongs is indicated in (**c**), and is consistent across (**b**) and (**c**). Pallium, Pal; subpallium, Sp; thalamus, Th; habenula, Hb; pretectum, Pt; tectum, Tec; tegmentum, Tg; cerebellum, Cb; and hindbrain, HB.

stronger functional relationships between the tectum and other regions, and with a greater number of highly correlated edges from the hindbrain to other regions (Supplementary Fig. 9). This pattern of findings, in turn, echoes observations from Fig. 4, which suggests that an uncoupling of spatially distinct perceptual and downstream networks drives habituation.

**The _fmr1_⁻/⁻ habituating graph is less stable and maintains more complex connectivity**. While we have shown that pairwise connections and functional identity are important during habituation, we next asked whether the higher-order structure of the evolving graphs also plays a role. We took a two-pronged approach for analyzing higher-order structure quantitatively in the correlation graphs: first, we characterized the dense connectivity patterns using dynamic community detection, and second, we detected sparse areas using persistent homology.

To understand the structure of densely connected areas of the graphs, we performed dynamic community detection, which probes how functional clusters in graphs change over time[80–84]. Briefly, this method first establishes communities of nodes based on their connectivity at each time point, and then tracks the community changes of each node through time. In this method, a structural resolution parameter γ and a temporal resolution parameter ω are used to define the number of communities and their tendency to change at different time points (Fig. 7a, see "Methods" section for details). Finding optimal γ and ω values is important as they can affect the results of the dynamic community metrics. Therefore, an assessment of a range of these parameters was first performed to find appropriate values (Supplementary Fig. 10 and see "Methods" section). Then, using the dynamic community detection approach, we found that the _fmr1_ mutant communities are generally less stable (presented more changes than WT communities), but specifically that the strongly habituating nodes (green functional cluster) show significantly higher flexibility (more community changes through time) (Fig. 7b, c). These strongly habituating nodes have higher cohesion (tending to change together with other nodes from the same community) and present higher promiscuity (tending to participate in a greater number of different communities through time) than in the WT graph (Fig. 7b, c). This pattern of findings indicates an increased involvement of this part of the graph during loom habituation in _fmr1_ mutants compared to WT. Interestingly, the weakly habituating nodes, and some of the moderately habituating nodes, show less cohesion in _fmr1_ mutants than WT, suggesting that the core visual pathway of _fmr1_ mutants may lose coordination and structure. Furthermore, the areas of the brain more affected by these alterations in flexibility, cohesion, and promiscuity are the subpallium, midbrain visual structures, and hindbrain (Fig. 7c–f). Altogether, these data suggest that the generally increased connectivity results in a less structured network, particularly in the core visual pathway, which then fails to uncouple elements of the secondary

processing (green nodes) as habituation occurs. This less stable and less structured network may lead to slower habituation because the elements producing the behaviors and further processing (green nodes) remain partially engaged.

To complement this assessment of the densely connected areas of the graph, we performed a second analysis, this time focusing on the sparsest areas. Specifically, we compared the sparsity patterns of the _fmr1_ mutant and WT graphs by analyzing the correlation matrices from each loom using persistent homology, a tool from applied topology that detects topological cavities or voids within weighted graphs (Fig. 7g, h)[85–88]. Briefly, in this analysis, the edges of a graph are added one by one, from strongest to weakest, until all of the connections are included, which creates a sequence of binary graphs. As we add edges, we can track the birth, evolution, and death of topological cavities (in dimensions 0, 1, or 2) within the sequence of binary graphs, which tells us about the lifespan of each persistent void in the original weighted graphs (Fig. 7h). The number of dimension 0 cavities counts the number of connected components, the number of dimension 1 cavities counts the number of voids surrounded by loops of four edges or more, and the number of dimension 2 cavities counts the number of void-enclosing shells formed from triangles in the graph. Persistent homology reveals that the _fmr1_ mutants' networks often contain more topological voids than WT networks, which suggests that the _fmr1_ mutant networks are noisier and less structured than the WT networks (Fig. 7i, j and Supplementary Fig. 11a–e). This idea derives from previous studies which shown that in weighted graph models, random graphs have been observed to have higher lifetime sums than do graph models that contain constrained organization[89–91]. Given this intuition, the consistent positive difference in lifetime sums between the _fmr1_ mutant persistent homology and WT persistent homology supports the community detection findings that the _fmr1_ mutant networks are more disordered than those of their WT counterparts.

Together, the analyses from our two complementary approaches indicate that _fmr1_ mutant animals have overly connected networks, which are less stable (present more community changes through time) and are noisier (suggested by a higher lifetime of voids), leading to altered function in the core visual pathway, more persistent coupling of secondary structures as loom habituation occurs, and a resulting slowing of habituation. Our results align with previous findings that showed functional connectivity alterations leading to more disordered network activity in _Fmr1_-null mice[92,93] which, like our zebrafish larvae, show impeded behavioral visual habituation[94]. Interestingly, previous calcium imaging and electrophysiological studies of somatosensory and visual cortex in _Fmr1_-mutant mice have also found more strongly correlated activity[95,96]. This suggests that stronger correlations across the nervous systems of FXS models is a generalized phenotype that could explain hypersensitivity to various stimuli.

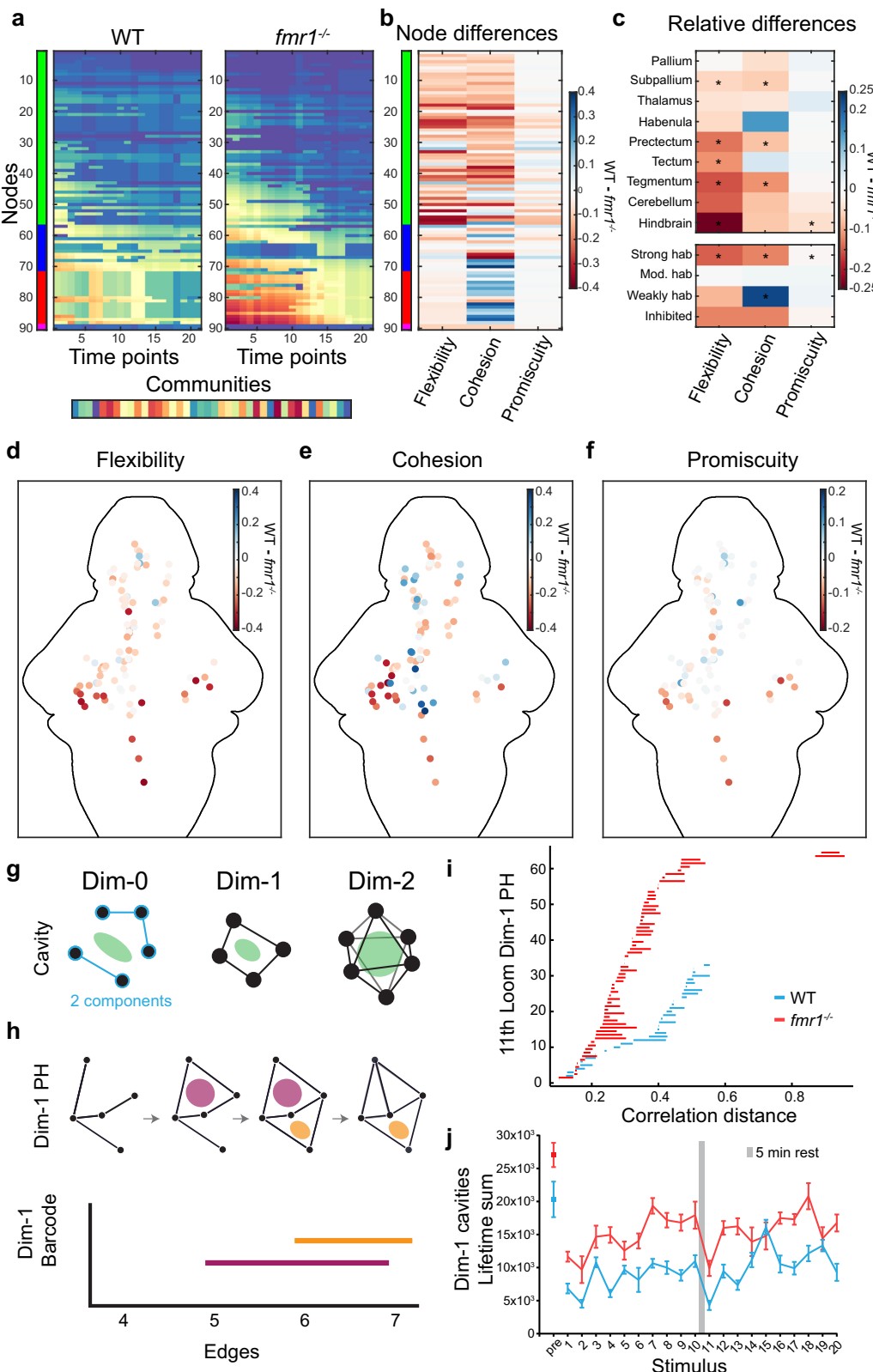

## Discussion

**A brain-wide model of visual habituation**. From an anatomical perspective, the core loom perception circuit can be inferred from the edges that remain active through habituation. These include edges within and among the tectum, thalamus, and pallium (Fig. 4b). The absence of habituation in these edges suggests that they are involved in perceiving a looming stimulus, and that they

are upstream of the sensorimotor transformation that controls behavioral outputs. The regions most affected during habituation (especially the hindbrain, but also including a subset of ROIs in the thalamus and the pallium) are likely downstream of this transformation.

The tectum is an important recipient of loom information[40–43,46,64,97], and communicates in different ways

**Fig. 7 Dynamic community detection and persistent homology across WT and _fmr1_⁻/⁻ graphs. a** Example community detection results obtained with $\gamma = 1.6$ and $\omega = 0.9$. **b** Relative (WT minus _fmr1_⁻/⁻) values of flexibility, cohesion, and promiscuity for each of the nodes. **c** Relative values of flexibility, cohesion, and promiscuity for 9 brain regions and by functional cluster. The color represents the difference of the median (WT minus _fmr1_⁻/⁻) and * indicates statistical significance ($p < 0.00384$) for a Friedman's test (one-sided) and Bonferroni correction. Details can be found in Supplementary Table 3. **d–f** Heat map of the relative (WT minus _fmr1_⁻/⁻) values of flexibility (**d**), cohesion (**e**), and promiscuity (**f**) for individual nodes across the brain. **g** Conceptual examples of structures that can be analyzed with the persistent homology method. **h** Schematic example of a persistent homology analysis. Persistent homology tracks cavities (pink and orange regions) across a sequence of networks in which edges are added according to their decreasing correlation strength (top), and the lifespans of these cavities can be represented as edges are added (bottom). **i** Example dimension 1 barcode graphs for _fmr1_ mutants and WT at the 11th loom. **j** Lifetime sums in dimension 1 of _fmr1_ mutants (red) and WT (blue) at pre-loom and 20 loom time points. Results for dimensions 0 and 2 are shown in Supplementary Fig. 11a–e. Centre represents the mean and error bars indicate 95% CIs.

with a variety of brain regions, making it an intriguing pivot point in the overall network. Those connections include non-habituating correlations with the pallium and likely outputs to the hindbrain that habituate strongly (Supplementary Fig. 9). The tectum also contains a high density of moderately habituating ROIs (Fig. 2h), whose activity most closely mirrors free-swimming behavioral habituation (Supplementary Fig. 6). These two observations raise the possibility that circuits within the tectum are responsible for the key changes in the sensorimotor transformation that produce habituation. This idea is reinforced by the drops in the correlation between moderately habituating ROIs and weakly habituating ROIs (blue-red edges) and between moderately habituating and strongly habituating ROIs (blue-green edges) during habituation. We propose a mechanism by which moderately habituating neurons in the tectum could uncouple the core visual circuit of weakly habituating (red) neurons from downstream circuits as habituation proceeds. These uncoupled circuits, principally comprising strongly habituating (green) ROIs, show interesting diversity reflective of distinct impacts that novelty and saliency play in different brain regions. The hindbrain's strongly habituating ROIs are more likely to correlate with the animal's actual escape responses (Supplementary Fig. 11f–h), as are those in other motor-associated regions including the cerebellum, pretectum, thalamus, and tegmentum. This suggests an interaction with these regions' premotor and motor circuits[63,64] and an acute role in escape. Other strongly habituating ROIs that uncouple from the tectum occupy the pallium, including the Dm, a fear processing area[65,98], and these are less likely to correlate to behavior on a trial-by-trial basis (Supplementary Fig. 11f–h), suggesting reduced higher-order representations of threat during habituation that are independent of trial-by-trial escape. The overall interpretation is that habituation involves the uncoupling of various downstream elements from visual perception circuitry, and implicates the tectum as the likely switch for this sensorimotor transformation.

Overall, we have shown that neurons with distinct habituating profiles to repetitive visual stimuli are present throughout the brain, and that the detailed responses of these categories of neurons can be modulated by the saliency and temporal details of the stimuli. These response profiles, viewed brain-wide at cellular resolution, reflect the rates of behavioral habituation to repeated looms, providing a framework for understanding the brain-wide network changes that mediate habituation. Using graph theory, we have shown that behavioral habituation tracks with a functional disconnection of a principally visual circuit in the fore- and midbrain, and of a response circuit that includes known premotor regions located in the hindbrain and higher-order forebrain regions that represent threats. The central location of the tectum (homologous to the mammalian superior colliculus) in this functional network, and the prominence of moderately habituating tectal neurons whose activity reflects behavioral habituation rates, suggest that this region is involved in visual learning. Given these properties, the tectum could serve as a pivot point for the sensorimotor transformation, a role that may be

conserved in birds and primates[99]. We have shown that this overall network is present in _fmr1_⁻/⁻ animals, but that its dynamics are shifted toward higher network correlations, greater transmission from sensory to premotor regions, and ultimately slower behavioral habituation (Fig. 5a). The _fmr1_⁻/⁻ networks also appear more unstable and less structured than the WT networks; these observations may be explained by an increase in persistent communication between the brain regions and functional clusters that would normally disconnect during habituation. These observations provide a brain-wide mechanism for slower sensorimotor learning that reflects previously reported behavioral phenomena in animal models and humans with FXS[56,78,79,94]. Importantly, it provides a departure point for targeted explorations of the circuit-level causes of learning and sensorimotor deficits in FXS and related psychiatric conditions.

## Methods

**Animals**. All zebrafish (_Danio rerio_) work complied with all relevant ethical regulations for animal testing and research in accordance with The University of Queensland Animal Welfare Unit and ethics approval SBMS/378/16. Adults were reared and maintained in a Tecniplast zebrafish housing system under standard conditions using the rotifer polyculture method for early feeding 5–9 days post fertilization. For the visual habituation experiments with different stimulus trains, we used _nacre_ zebrafish embryos of the TL strain expressing the transgene, _elavl3:H2B-GCaMP6s_[100]. For the _fmr1_ experiments, zebrafish embryos were bred by incrossing zebrafish heterozygous for _fmr1_[hu2787] [77] and _elavl3:H2B-GCaMP6s_, to produce clutches with a 1:2:1 Mendelian ratio (wild type: heterozygous: homozygous) for _fmr1_[hu2787]. The _fmr1_[hu2787] mutants have a change (C to T) in the _fmr1_ coding region leading to a nonsense-mediated decay and the loss of the protein[77]. Following the experiments, larvae were genotyped as previously described[74]. All fish were produced by natural spawning and reared in Petri dishes with embryo medium (1.37 mM NaCl, 53.65 μM KCl, 2.54 μM Na₂HPO₄, 4.41 μM KH₂PO₄, 0.13 mM CaCl₂, 0.16 mM MgSO₄, and 0.43 mM NaHCO₃ at pH 7.2) at 28.5 °C on a 14-h light: 10-h dark cycle.

**Stimulus train for behavioral experiments**. The stimulus train consisted of three blocks of 10 looms with 5 min of rest (with a white screen) between each block. The loom was initiated with a dot that started expanding after 1 s. The minimum angle of the loom was ~11° and the maximum angle of the loom was ~90°. The fast looms reached their maximum angle in 2 s and the slow looms in 4 s. This was followed by 2 s of black screen and a 9 s slow fade back to white, designed to avoid any neural OFF responses. The screen remained white until the next loom initiation for a variable duration depending on the desired inter-stimulus intervals (ISI) of 18, 20, or 22 s for the f20 and s20 paradigms and 54,60, or 66 s for f60 and s60. A sound stimulus of 300 Hz at ~85 dB was played 3 times for 1 s with 1 s ISI. The first presentation was 25 s before the 21st loom. The video and sound were displayed by a monitor (10.1 1366 × 768 Display IPS + Speakers - HDMI/VGA/NTSC/PAL, Little Bird, Australia). Since the sound stimulus did not produce any marked dishabituation, in spite of eliciting calcium responses, we did not analyze this aspect of the experiment.

**Behavioral experiments**. Individual 6 dpf larvae were placed in each well of the 12-well arena (circular plugs of agar were removed to produce the wells). The wells were filled with embryo medium and were placed at 1 cm above a screen inside a dark chamber, and all larvae received the same stimulus train. The chamber was kept in the dark but was illuminated with infrared LEDs. A Basler acA1920 camera recorded the movements from above, a lens (40 mm Thorlabs) and a 665 nm longpass filter (FGL665 - Ø25 mm RG665 Colored Glass Filter, Thorlabs) delivered infrared light to the camera with a weak signal from the screen that confirmed the timing of the looming stimuli. Movies were recorded using the Basler Video Recording Software (v1.3, Basler AG). Movements were tracked in bins of 1 s using

the zebrafish tracking Viewpoint software (v1.4, ZebraLab, ViewPoint Life Sciences, France), tracking three-speed categories: <0.5 mm/s, 0.5–30 mm/s, and >30 mm/s. The output of the tracking was then analyzed using a Matlab script. Escape responses were defined as one or more movements above 30 mm/s during a loom presentation. Further statistical analysis and graphs were made in GraphPad Prism v8.3.1 and R 3.5.1[101]. The sound failed to produce a clear dishabituation so this effect was not further analyzed. The fitted curves were done in GraphPad Prism v8.3.1 with the exponential one-phase decay curve from the 1st to the 10th loom of each block, using a Least Squares regression and plateau to 0. We used the lme4 and MuMIn R packages to generate the GLMM and to calculate the $R^2$. The model was fitted for a binomial distribution with the formula: response = loom + speed + ISI + (1/fishID).

For the *fmr1* experiments, the procedures were the same, however, the stimulus train was a shorter version of the s20 with 20 looms instead of 30, as we did not observe an effect of the auditory tone between the 20th and the 21st loom. When the experiment ended, larvae were processed for genotyping. The quantification of the data was performed blind to the genotype for the *fmr1* experiments. The binomial test was performed one-sided with the escape responses of the *fmr1* or Het larvae in each loom versus the probability of response of the WT for that same loom. All the statistical tests for the behavioral analysis assumed repeated measurements (for the multiple loom presentations) and non-normal distributions.

**Sample preparation for calcium imaging**. Imaging was performed on 6 dpf larvae that were embedded upright in 2% low melting point agarose (Sigma, A9045) and transferred to a 3D-printed imaging chamber[102]. Imaging chambers were filled with embryo medium once the agarose had set and the tail was freed[103] so that escape responses could be monitored. The imaging chamber was composed of a 3D-printed base (24 × 24 mm) with four posts (3 × 3 × 20 mm) raised along the four corners of the platform. The four outward faces of the chamber were fixed with a glass coverslip (20 × 20 mm, 0.13–0.16 mm thick). A glass window on the bottom of the chamber allowed filming of tail movements[102]. For the *fmr1* experiments, larvae were processed for genotyping when the experiment ended.

**Loom stimulus train for calcium imaging**. Looms were presented on a 75 × 55 mm LCD generic PnP monitor (1024 × 768 pixels, 85 Hz, 32-bit true color) with a NVIDIA GeForce GTX 970 graphics card. The monitor was positioned 30 mm to the right of the larvae, and was covered by a colored-glass alternative filter (Newport, 65CGA-550) with a cut-on wavelength of 550 nm. The minimum angle of the loom was ~10° and the maximum angle the loom covered was ~82°. The auditory stimulation (a 100 Hz sound at 100 dB before the 21st loom) was presented with two audio speakers (Logitech Z213) placed at ~20 cm from the fish. The background noise level was 40 dB. As for the behavioral experiment, in the *fmr1* experiments, the procedures were the same but with a shorter version of the s20 stimulus train.

**Microscopy**. Zebrafish larvae, individually mounted in the imaging chamber, were imaged for *elavl3:H2B-GCaMP6s* on a custom-built SPIM microscope[102,104]. To avoid stimulating the eyes with the light sheet, the side laser path of the SPIM was blocked, and the front SPIM plane was restricted to a space between the eyes using a vertical aperture. Micro-Manager (v1.4.22) was used to capture images, which were binned 4 times to a final resolution of 640 × 540 pixels at 16-bit in tagged image file (TIFF) format. Fifty horizontal sections at 5 µm increments were captured and imaged at 2 Hz. Recording of the brain activity started 30 s before the first stimulus onset and stopped after the return to white from the last loom of each block, resulting in three separated acquisitions. To image the larva and record its tail movements, a 4 × 0.1NA Olympus microscope objective (PLN 4X) was placed below the sample chamber[105], coupled with a tube lens projecting the image onto a Basler acA1920 camera, recording at 30 fps with the Pylon Viewer (v5.2.0, Basler AG) software.

At the end of each experiment, a single high-definition scan of non-binned images was recorded with 100 ms exposure time and 2 µm increments to be used for the registration of the brain of each fish (see below).

**Analysis of calcium imaging data**. Calcium imaging data from the three acquisitions were concatenated in ImageJ v1.52c as a combined time series and then separated into individual slices (50 planes per fish). Motion correction was performed using Non-Rigid Motion Correction (NoRMCorre) algorithm[106], and fluorescence traces were extracted and demixed from the time series using the CaImAn package (version 0.9)[107,108] (http://github.com/flatironinstitute/CaImAn). We used 4000 components per slice to ensure that we would not miss any ROIs during the initialization step of CaImAn. The risk of over-segmentation was mitigated by a merge step using a threshold of 0.8 to merge overlapping ROIs. The order of the autoregressive model was set at 1 to account for the decay of the fluorescence, our acquisition speed being too slow to account for the rise time. The gSig (half-size of neurons) was set at 2, based on estimates of the sizes of the nuclei in our images. We did not use any temporal or spatial downsampling and the initialization method was 'greedy_roi'. The components were updated before and after the merge steps, empty components were discarded, and the components were ranked for fitness as described previously[107].

**Analysis of whole-brain activity data**. For the experiment with four stimulus trains, the resulting ROIs and fluorescent traces from the CaImAn package were pooled from larvae of each stimulus train (*n* for the 4 data sets: f20 = 11, f60 = 8, s20 = 10, s60 = 10), and then z-scored per data set. For the analysis of the change in response based on the first loom (Fig. 2a–e), a linear regression was performed, using a stereotypical GCaMP6s trace as regressor, to detect the responses to the first loom. The ROIs with $r^2 > 0.5$ were then selected to see the intensity of response at the 2nd and 10th loom presentations. A *K*-means clustering by cityblock distance with 50 components and 5 replicates was done for each data set with the denoised output from CaImAn and was also run a second time with noise added back to allow for the detection of negative responses[109]. Another *K*-means clustering was performed with 200 components to test whether additional subtypes of responses could be discovered and to confirm representation of common clusters across the data sets, but this resulted in numerous clusters drawn from individual fish, which we took as a sign of overclustering. From the 50 clusters for each data set, a subset was manually selected based on their responses to the looms and sound and their general presence across all data sets and individual fish. To select representative visual habituating clusters, we set as a criterion that they had to be present in all 4 data sets and in a majority of fish (>80% in each data set). This resulted in the weakly habituating, moderately habituating and strongly habituating clusters. Other potentially interesting clusters were found in multiple, but not all four data sets. To include some of these for further analysis, our selection criteria were that these clusters had to be confirmed in both f20 and s20 in the *K*-means with 200 clusters, their distributions in the brain were similar across both data sets, and that they were present in >90% of their fish. These criteria led to the inclusion of strongly habituating subtypes, the sound-responsive cluster, and inhibited responses. These clusters (three strongly habituating, a moderately habituating, a weakly habituating, an inhibited, and a sound-responsive cluster) were used as regressors for subsequent analyses of the four data sets. All ROIs from each of the 4 data sets were modeled by linear regression to each of these regressors. As the 60 s ISI time series were longer, the time series were trimmed around the 30 looms to perform the linear regression. ROIs with an $r^2$ value higher than 0.3 were selected for further analysis. The selected ROIs were categorized by correlation to each of the 7 selected regressors. The auditory cluster was not analyzed after this point. After filtering the ROIs with the linear regression, all of the clusters were found in all the fish of each data set except for the inhibited cluster, which varied in representation across data sets (f20 = 81%, f60 = 87%, s20 = 100%, and s60 = 63%). We confirmed that the clusters could be found in most or all larvae, but 3 fish (1 from f20 and 2 from f60) were discarded because their contribution to one of the habituating clusters was above 50% of the total number of ROIs for that cluster, so they were deemed as outliers in terms of responsiveness. To find the motor evoked calcium responses, we first used ImageJ to detect the tail movements from the behavioral imaging. We used a polygon ROI covering half of the tail to extract the mean gray values of the time series. Substantial tail movements produced large peaks and were flagged as movement events. We then build regressors for individual larvae inserting a stereotypical GCamp6s trace to the movement timing for each larva. Finally, we used a linear regression with the motor regressor of each larva as for the habituating clusters, and selected ROIs with an $r^2$ value higher than 0.2.

For the t-SNE[110] (Supplementary Fig. 4a) we used the Matlab function with a correlation-based distance and the following parameters: Perplexity = 184, Exaggeration = 40, Iterations = 3000. For further analysis, we pooled together the three strongly habituating clusters and we excluded the sound response cluster, resulting in four main clusters.

To calculate the proportions of ROIs for a given cluster that appear in each brain region (Fig. 3b), the number of ROIs of each cluster in each brain region was divided by the total number of ROIs of that cluster in the whole brain. We did this for each individual larva, created a mean for each data set, and then averaged these values across all four data sets. The same procedure was used to calculate the proportion of each cluster within all loom-responsive ROIs per brain region (Fig. 3c).

For the correlation analysis of habituation dynamics between the free-swimming behavior and brain responses, we performed a Pearson correlation between the normalized responses of each fish and the free-swimming probability responses of the matching group. Then the correlations were averaged by brain region and cluster subtype with the requirement that at least three fish were contributing to each combination. Finally, these values were averaged when the four groups fulfilled the previous requirement (Supplementary Fig. 6c). For the analysis of the normalized responses in the tectum (Supplementary Fig. 6d), a mean of the tectal ROIs' responses for each cluster was calculated for each individual fish, then the maximum response per loom was calculated based on the maximum z-score value in the window of the loom presentation adjusted by the baseline before each loom. These values were normalized to the first loom response, and a mean of the normalized maximum response was calculated for each data set. To analyze the differences between stimulus trains in moderately habituating responses in the tectum, we performed a nonlinear regression fitting a one-phase decay curve with a least-squares regression as a fitting method. We tested for differences in Plateau and rate constant values (*K*) with the constraint that *K* must be greater than 0. The curve was fitted only from loom 1. To test the amplitude of recovery, we used a mixed-effects model and a Sidak's multiple comparisons test between the 10th and the 11th looms. To compare the tectal responses of the

strongly habituating, moderately habituating, and weakly habituating normalized responses with the matching free-swimming behavioral results, we calculated the Pearson correlations coefficients between the responses of each fish and the free-swimming responses of its matching stimulus train. We then averaged these results to compare across habituating profiles.

To locate the subset of strongly habituating neurons that are involved in motor behaviors (Supplementary Fig. 11f–h), we calculated the Spearman correlation coefficient between each strongly habituating ROI from the f20 data set and the motor regressor of its respective fish. We then selected the ROIs above a correlation coefficient of 0.3066 (the mean, 0.1522, plus one SD, 0.1544). Finally, we calculated their proportion compared to the strongly habituating ROIs of each of the brain regions previously analyzed.

For *fmr1* experiments, we performed a k-means with 50 components with the traces of all the fish. We then selected 8 clusters based on their possible loom responses. Then we performed a linear regression and selected the ROIs with an $r^2$ value above 0.3. As their location and average calcium traces were similar to the functional clusters previously found, we classified the ROIs into the functional clusters from our original s20 data set using correlation as described above. All data were quantified blind to genotype. For Fig. 5b, we chose a random sample ($n = 11$) of Hets to match WT ($n = 10$) and $fmr1^{-/-}$ ($n = 11$).

The analysis was done using Matlab R2018b and GraphPad Prism v8.3.1.

**Construction and validation of correlation matrices and graphs**. To allow formal statistical comparisons between individual fish and across groups in the context of our graph theoretical analysis, we had to cluster our 144,709 responsive ROIs spatially while preserving their functional identity. This approach permitted us to have comparable anatomical and functional nodes across all individual fish and groups. To do so, we performed *k*-means clustering on the 3-dimensional spatial coordinates of the ROIs[111] of each functional cluster, in each brain region, with *k* number of clusters. The value of *k* was chosen based on the number of ROIs. For regions with fewer than 200 ROIs, no node was placed; between 200 and 500, 1 node; between 500 and 1000, 2 nodes; between 1000 and 3000, 3 nodes; and >3000, 4 nodes. This node attribution was intended to strike a balance between (i) including relatively sparse populations that may, nonetheless, make functional contributions, and (ii) weighting our analysis to some degree toward more abundant response types. This method produced 102 nodes, but we discarded three nodes that had three or fewer fish contributing to them. For the remaining 99 nodes, we computed the cross-correlation between the mean loom response of their ROIs and generated individual matrices for each larva, and each loom presentation. We then averaged the matrices of each data set across larvae to produce a single set of nodes across the brain, each with averaged correlations during the relevant stimulus train. This approach permitted apples-to-apples comparisons of brain-wide responses across groups (receiving different stimulus trains or with different genotypes).

To assure that downsampling our responses to 99 nodes did not eliminate key properties of the network, we performed a sensitivity analysis with a range of node numbers. Specifically, we assessed the density and participation coefficient metrics in graphs where we had doubled or quadrupled the number of nodes. We also performed a graph analysis of all the ROIs of each fish. All the individual ROIs' signals were cross-correlated and we generated correlation matrices for each loom presentation as we did in our node-based approach, we then quantified density and the participation coefficients (Supplementary Fig. 7). The results of this sensitivity analysis show that our 99 node approach provides similar dynamics to those found with a higher number or nodes or all ROIs in individual fish.

To further validate the results from our graph model, we compared them to results generated from null models, using the f20 data set. We used the amplitude-adjusted Fourier transform (AAFT) to generate surrogates of the time series. Unlike a random surrogate model, this method allowed us to preserve features of the original time series, like the mean, variance, and amplitude distribution, thereby increasing the null model's stringency[112]. To generate such null models, instead of averaging the individual correlation matrices as above, we first averaged the time series of each node across all f20 fish (Supplementary Fig. 8a, left). Then, using an AAFT algorithm[113], we generated a first model with surrogates of each node's time series (Supplementary Fig. 8a, middle), and a second model with surrogates of each node's time series within the loom time windows used for the correlation analysis (Supplementary Fig. 8a, right). We then generated the correlation matrices as above.

As a final validation of our results, we used the f20 dataset to perform a leave-one-out cross-validation, to ensure that no single fish drove our overall results. We generated group-averaged matrices as before, but we systematically excluded one fish from the average each time. Five examples of these matrices are shown in Supplementary Fig. 8e, closely resembling the data from our entire group of fish(Supplementary Fig. 8f). The same approach was also used to compare the number of edges >0.75 between WT and *fmr1*$^{-/-}$ larvae (Fig. 6a).

**Quantitative analyses of graphs**. Having validated our matrices, we proceeded with an array of quantitative analyses. First, we aimed to measure the overall similarity between pairs of graphs as a means of gauging the completeness of recovery after a break in the stimulus (Fig. 4). To identify the graph most similar to the 11th trial of the f20 and f60 data sets, we calculated the correlation between the

matrices for the first 10 and the matrix for the 11th loom for each stimulus train. To perform this correlation, we used a vector composed of all elements in the upper right half of the matrix (above the diagonal), since each matrix is a mirror image across this diagonal. We then identified the trial number with the highest Pearson correlation coefficient to the 11th loom, when recovery takes place.

For further quantitative analyses of the graphs' metrics, we used the Brain Connectivity Toolbox[114]. We first generated weighted connectivity matrices and filtered out edges with an absolute correlation value below 0.75. We then calculated the graph density (ratio of the present edges to the total possible edges of the graph) and the node-specific participation coefficient[115], defined as:

$$P_i = 1 - \sum_{k=1}^{K} \left(\frac{S_{iC_k}}{S_i}\right)^2 \tag{1}$$

where $S_{iC_k}$ is the total edge weight of node $i$ to nodes in community $C_k$ and $S_i$ is the total number of edges of $i$. The participation coefficient was calculated by letting the four functional clusters identified previously (strongly habituating, moderately habituating, weakly habituating, and inhibited) represent the partition ($K$).

The *fmr1* data set was treated similarly using the spatial nodes from the previous data set. ROIs were assigned to each node based on the smallest Euclidian distance. After discarding nodes represented in fewer than 3 larvae, we ended up with 90 nodes for this analysis. As before, we calculated the correlation between time series pairs, and generated individual fish matrices for each loom presentation.

**Dynamic community detection**. The multilayer graphs and dynamic community detection is based on previous work[80] and was performed on the unthresholded matrices using the MATLAB genlouvain.m function from the GenLouvain v2.2 toolbox[116]. The multilayer modularity quality function is given as follows:

$$Q_{\text{multilayer}} = \frac{1}{2\mu} \sum_{ijlr} \left\{ \left( A_{ijl} - \gamma_l P_{ijl} \right) \delta_{lr} + \delta_{ij} \omega_{jlr} \right\} \delta(C_{il}, C_{jr}) \tag{2}$$

where $\mu$ is the total edge weight, $A$ is an adjacency matrix and $A_{ijl}$ is its $ij$th element at layer $l$. The element $P_{ijl}$ gives the expected weight connecting node $i$ and node $j$ under a null model at layer $l$ and $\delta$ is the Kronecker delta. The partition in a number of communities at each time point was determined by the structural resolution parameter $\gamma$. Smaller values of $\gamma$ will generate fewer communities, while higher values will increase the number of communities. The temporal resolution parameter $\omega$ determines the strength of the connections between the nodes of different time points, influencing the rate of community change. Low $\omega$ values produce dynamic graphs with a high tendency to change while high values generate more time rigid community partitions.

To find the optimal $\gamma$ and $\omega$ for our multilayer community detection, we used a combination of two approaches. The first approach involved finding the optimized maximization of the modularity quality function ($Q$), which are the $Q$ values that differ the most from a null model and have the smallest variability. In this case, we used a comparison against a temporal null model[112]. The second approach is bounding the $\gamma$ and $\omega$ parameters to ensure that the community detection results are informative. The goal was to identify a $\gamma$ value that produced an appropriate number of communities and a $\omega$ value that was neither too rigid nor too dynamic.

For the first approach, we used the multilayer graph of each genotype and performed 100 repetitions of the maximization of the modularity quality function ($Q$) in a wide range of $\gamma$ and $\omega$ values (0.1–2.5 and 0.1–2, respectively). We then calculated the mean and variance of $Q$ for each combination of parameters generating matrices of these values. This same procedure was performed for a temporal null model of each genotype in which the time points were randomly permuted. The averaged $Q$ values of the original graph were subtracted by the respective averaged $Q$ values of its temporal null model. These values were then multiplied by the relative variance [-var-max(var)] to find the optimized $Q$ values at which the greatest difference from the null model and the minimum variance across repetitions was observed. We then computed the average optimized $Q$ for the 3 genotypes data sets.

For the second approach, we performed the community detection 100 times and then found the representative partitions using the Consensus Iterative.m function[112] for the previous range of $\gamma$ and $\omega$ values (0.1–2.5 and 0.1–2, respectively). We then established the following rules. Using the WT data set as a reference, we did not include combinations of parameters that had more than 60 or fewer than 4 communities. This first rule limited the lower and higher range of $\gamma$ values. We also set values such that at least a third of the nodes (30 for these data sets) would change community between the pre-loom measurement and the first loom and also between the 10th and the 11th loom. This rule established the higher limits of the $\omega$ values. Finally, the consensus communities obtained with the 34 pairs of $\gamma$ and $\omega$ values that had optimized $Q$ values above the mean, and that respected our dynamic community rules, were used for the analysis (Supplementary Fig. 10).

To describe the roles of each node in their graphs, we used the previously described community measures of flexibility, cohesion, and promiscuity[84], which are available online (http://commdetect.weebly.com/). The flexibility coefficient is a simple yet important metric as it indicates the number of times a node changes

community normalized by the total possible changes[81], represented by the formula

$$\xi_i = \frac{g_i}{L-1} \quad (3)$$

where $L$ is the number of layers and $g$ the number of times a node changes community.

The cohesion strength indicates the degree to which a node tends to change communities mutually with another node. This is defined as

$$\Omega_i = \sum_{j \neq i} M_{ij} \quad (4)$$

where $M$ is a cohesion matrix of edge weights indicating the ratio of times a pair of nodes moves to the same community together:

$$M_{ij} = \frac{g_{ij}^{\mathrm{mut}}}{L-1} \quad (5)$$

Finally, the promiscuity measure is calculated based on the fraction of communities to which the nodes belong across all time points[117]:

$$\Psi_i = \frac{g_i^{\mathrm{dif}}}{K-1} \quad (6)$$

where $K$ is the total number of communities and $g_i^{\mathrm{dif}}$ is the number of changes to new communities of node $i$. Therefore, this metric is relevant to determine if a node with high flexibility is just changing between a few possible communities or if it is actually joining a wide range of them.

These measures were calculated for the selected combinations of γ and ω for the data set of each genotype and the results were analyzed using the MATLAB functions friedman and multcompare. As we assumed non-normal distribution, a Friedman's test was performed followed by a multiple comparisons test of the average column ranks between the results of the 3 genotypes using a Bonferroni adjusted alpha for those comparisons. To select the significance values we applied a Bonferroni correction for the 13 tests (9 brain regions and 4 cluster types) and looked for $p < 0.05/13 = 0.0038$. The results can be seen in the Supplementary Table 3.

**Topological analysis**. To identify topological differences between *fmr1* and WT fish, we used the leave-one-out approach to generate one 90 × 90 correlation matrix for each loom and each fish left out. We computed the persistent homology in dimensions 0 through 2, of the filtered clique complex of each correlation matrix, using the open-source Eirene package[118]. We set the correlation value as the filtration parameter. The output of the persistent homology calculation is a barcode in which each bar corresponds to a persistent cavity, and the bar spans from the persistent cavity birth (the highest correlation value in which the cavity exists) to the death (the correlation value at which the persistent cavity is tessellated). The absolute value of the difference between the death and birth values is called the persistent cavity lifetime. Summing over all persistent cavities in dimension n of a barcode yields the lifetime sum. Please see refs. [88,119,120] for more details on the mathematics of persistent homology.

**Registration to a reference brain**. We used Advanced Normalization Tools (ANTs, https://github.com/ANTsX/ANTs) to register our results on the H2B-RFP reference of Zbrain[121–123]. The high-definition stacks were used to build a common template, before registering this template to the Zbrain atlas[102]. The resulting warps were sequentially applied to the centroids of extracted ROIs to map them all in the same frame of reference. The Warped ROI coordinates were then placed in each of the 294 brain regions defined in the Zbrain atlas[123].

**Data visualization**. We used Unity to represent each ROI centroid as a sphere. Their diameter was adjusted based on the number of ROIs to be able to visualize the different clusters (Strongly habituating = 2; Moderately habituating = 3; Weakly habituating = 4; Inhibited = 6). An isosurface mesh of the zebrafish brain was generated from the Zbrain masks for the diencephalon, mesencephalon, rhombencephalon, telencephalon, and eyes using ImageVis3D[124]. The mesh was imported in Unity (v2019.3.0a2) and overlaid to the ROIs.

The colormaps used for Figs. 2, 4–7 and Supplementary Figs. 2, 4–8 were generated using two Matlab® functions: The cbrewer function, https://au.mathworks.com/matlabcentral/fileexchange/34087-cbrewer-colorbrewer-schemes-for-matlab (accessed in May 2019) which includes specifications and designs developed by Cynthia Brewer (http://colorbrewer.org/), and the MatPlotLib 2.0 default colormaps ported to Matlab, https://au.mathworks.com/matlabcentral/fileexchange/62729-matplotlib-2-0-colormaps-perceptually-uniform-and-beautiful (accessed in May 2019).

The circular graphs (Fig. 6 and Supplementary Fig. 9) were made with a modified version of the code from Matlab®'s circularGraph toolbox. https://www.mathworks.com/matlabcentral/fileexchange/48576-circulargraph/ (accessed in May 2019).

The density plot in Supplementary Fig. 4b was made with dscatter function. Made by Robert Henson and found in Flow Cytometry Data Reader and Visualization (https://www.mathworks.com/matlabcentral/fileexchange/8430-flow-cytometry-data-reader-and-visualization), MATLAB Central File Exchange. (accessed in November 2020).

Figures were produced using Matlab R2018b and GraphPad Prism v8.3.1 and assembled in Adobe Illustrator CS6.

**Reporting summary**. Further information on research design is available in the Nature Research Reporting Summary linked to this article.

## Data availability
Data used in this paper are stored in the University of Queensland's Research Data Manager repository, and are publicly available in the "MarquezLegorreta_et_al_2021_Datasets" database, available at https://doi.org/10.48610/9549fdc. Source data are provided with this paper.

## Code availability
All scripts can be found at the emarquezUQ/ZF_Loom_Habituation_MarquezLegorreta_et_al_2021 github repository (https://github.com/emarquezUQ/ZF_Loom_Habituation_MarquezLegorreta_et_al_2021).

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

## Acknowledgements

We thank Ulrike Siebeck for conceptual discussion about this project and Scott lab members for feedback on the manuscript. Support was provided by an NHMRC Project Grant (APP1066887), ARC Future Fellowship (FT110100887), a Simons Foundation Pilot Award (399432), a Simons Foundation Research Award (625793), and two ARC Discovery Project Grants (DP140102036 & DP110103612) to E.K.S. The research reported in this publication was supported by the National Institute of Neurological Disorders and Stroke of the National Institutes of Health under Award Number R01NS118406 to E.K.S. The content is solely the responsibility of the authors and does not necessarily represent the official views of the National Institutes of Health. Support was also provided by the Australian National Fabrication Facility (ANFF), QLD node, and by Support was provided by a Rebecca Cooper Medical Research Project Grant No PG2019405 and a NHMRC Emerging Leader Fellowship No. 1174145 to J.G. E.M.-L. gratefully acknowledges funding for postgraduate studies provided by the Mexican National Council for Science and Technology (CONACYT) and The University of Queensland. D.S.B. would like to acknowledge the Army Research Office (Bassett-W911NF-14-1-0679 and Grafton-W911NF-16-1-0474).

## Author contributions

E.M.-L. contributed to the conceptual design of the project, collected and analyzed most data, and contributed to writing the manuscript. L.C. collected *fmr1* data and performed genotyping. M.P. collected behavioral data for the *fmr1* experiment. I.A.F.-B. and M.A.T. built and maintained the light-sheet microscope. G.C.V. designed the data analysis pipeline for calcium imaging data and performed some data analyses. J.G. contributed consultation and computational resources for the behavioral experiments. E.S. contributed to the project's concept, guided analyses, and contributed to writing the manuscript. A.S.B and D.S.B. contributed computational consultation during the revision stages of the project, and edited the manuscript.

## Competing interests

The authors declare no competing interests.
