## [Peer Review File · Nature Communications]

REVIEWER COMMENTS

Reviewer #5 (Remarks to the Author):

This study by Marquez-Legorreta et al takes an impressive technical approach to address a long-standing question: how does an entire network adapt during learning? They use habituation to looming stimuli in larval zebrafish, first validating and comparing different stimulus presentation paradigms, and then applying whole-brain Ca²⁺ imaging to this problem. They further extend their analysis to global connectivity using graph-theory. They then perform these experiments in a learning-deficient model, *fmr1* mutants, and describe how these mutants show alterations in network-level effects. I found this to be a very interesting study.

My concerns/feedback are as follows:

Clustering method -- a very large number of clusters (50) were chosen for activity profiling, and then these appear to have been manually selected among and combined. Why is this method needed, rather than simply choosing a smaller number of clusters? It makes me wonder about the meaningfulness of these clusters that are arrived at in a somewhat arbitrary way. Since this is the basis of further graph network analysis, I am a bit concerned about how realistically this method captures the true diversity found in the response profiles. I would suggest that if the study used a method that does not rely on manual selection/pooling, this would strengthen this aspect of the study.

The network modeling results (Figure 4-7) are interesting. These are not methods I have personal experience with, but it seems to me that many of the panels are effectively showing the same thing with different display styles. For example are Figure 4a and b, and c and d, not showing the same data? I find all these figures a bit overwhelming, and expect that some readers may feel the same way. I would suggest that the authors reduce these figures to the main effect observed, displaying only one instance of the data, and moving the rest to supplementary figures.

My main concern with the graph analysis is how sensitive such an analysis is to biological noise, and therefore how meaningful changes observed are. Since the authors do not appear to have a negative control for such an effect, such as two independent populations of wt fish, it is difficult for me to tell whether the observed differences really gain more meaningful insight than the traditional correlation/clustering based methods in Figure 2 and 3.

Concerns with the *Fmr1* mutants. From my perspective, the data shown in Figure 5a is not convincing. Based on this data alone I would not be very confident that indeed these mutants exhibit a meaningful phenotype at the behavioural level. How many different clutches of larvae went into this experiment? Has this effect been validated in different biological replicates in separate clutches? The study does report a "significant" difference in individual timepoints, but considering the large sample sizes and noise inherent in behavioural data, I am not convinced that there is a meaningful difference here based only on a p-value. This of course makes me wonder about the meaningfulness of the graph-based analyses in these mutants (though these appear to be consistent with deficits in habituation). But without a relevant negative control for these analyses (see above) it is again difficult to know how much of this difference may be due to biological noise, especially because no significant changes in the cluster distributions was found (fig 5b)

Figure 1 - why not use color? Panel b is very confusing. Why is white and ISI not a constant number? What is the difference between white and ISI?

The use of the dishabituation stimulus. The authors conclude that this does not produce an effect. The potential reasons for this should be discussed. If there is no effect, then why do subsequent imaging experiments include this dishabituation stimulus?

I was specifically asked to comment on the Author's response to previous Reviewer 4 (R4). R4's first major concern was that of a lack of technical novelty. In my opinion, the study should not be judged based on what techniques it employs, but what biological insight is gained. I am satisfied with the response of the authors to this point.

R4's second major concern is summarized by their statement

"...the results are just descriptive. There is no clear unique insight into brain function that was gained from their analyses. Their results fit with established models of information flow in the brain. The authors did not attempt to reveal underlying mechanisms, for example through some perturbation of the key regions involved."

While this criticism is true, it could be equally applied to many neuroscience studies published in top journals (though maybe not for the zebrafish field, unfortunately for us!). The proposed manipulations would certainly be a huge boon to the study, but without a way to genetically target the relevant neurons, these studies are exceptionally challenging. The authors do in fact present a manipulation (the *frm1* mutants), which I feel is very interesting (though I do have concerns related to the presence of a meaningful behavioural effect in these mutants, described above).

Reviewer #5 (Remarks to the Author):

This study by Marquez-Legorreta et al takes an impressive technical approach to address a long-standing question: how does an entire network adapt during learning? They use habituation to looming stimuli in larval zebrafish, first validating and comparing different stimulus presentation paradigms, and then applying whole-brain Ca2+ imaging to this problem. They further extend their analysis to global connectivity using graph-theory. They then perform these experiments in a learning-deficient model, fmr1 mutants, and describe how these mutants show alterations in network-level effects. I found this to be a very interesting study.

My concerns/feedback are as follows:

Clustering method -- a very large number of clusters (50) were chosen for activity profiling, and then these appear to have been manually selected among and combined. Why is this method needed, rather than simply choosing a smaller number of clusters? It makes me wonder about the meaningfulness of these clusters that are arrived at in a somewhat arbitrary way. Since this is the basis of further graph network analysis, I am a bit concerned about how realistically this method captures the true diversity found in the response profiles. I would suggest that if the study used a method that does not rely on manual selection/pooling, this would strengthen this aspect of the study.

Comment #1:

Sorry, we could have been clearer about the aspects of this that were manual, and also about our rationale for starting with a large number of clusters before further selection. We tested a few different clustering approaches and concluded that k-means was the most appropriate to begin with, as it is relatively fast and it is (at least initially) unsupervised. Our rationale was that by clustering each dataset separately (f20, f60, s20 and s60), we could then compare across the results and retrieve the common functional clusters. When using k-means, it is necessary to choose a total number of clusters into which your data will converge. It is important to keep in mind that we ran this k-means on the full datasets, meaning that calcium traces could belong to any neuron in the brain, whether it was responsive to stimuli or not. We did this to be open to the possibility of finding more complex responses such as integration. This means, however, that thousands of traces have no relationship with our stimuli, and we needed a means of removing them before performing our further analyses. We tried a great variety of initial cluster numbers. Generally speaking, if we used a small number of clusters, we got very broad response profiles contained within the same cluster, restricting our ability to find variability and describe selective responses meaningfully. With an excessively large number of clusters, artifacts in individual animals (body movements, for instance) began to drive cluster selection (this is mentioned in lines 711-715). In the end, we went with a relatively large number of clusters (50), based on the logic that we could identify clusters that were meaningful and represented by most fish while still spotting and removing artifacts or the products of individual fishes' particular responses. We did this in an unbiased way, at least in the sense that we applied exclusion criteria to all clusters. Of course, we defined these exclusion criteria based on our empirical observations, so they're not truly unbiased, but we submit that they offer a reasonable balance between inclusion of real and specific signals with exclusion of artifacts. In the revised manuscript, we have added details about these exclusion criteria to the "Methods" section (lines 715-727), and no further manual selection of clusters was performed.

The network modeling results (Figure 4-7) are interesting. These are not methods I have personal experience with, but it seems to me that many of the panels are effectively showing the same thing with different display styles. For example are Figure 4a and b, and c and d, not showing the same data? I find all these figures a bit overwhelming, and expect that some readers may feel the same way. I would suggest that the authors reduce these figures to the main effect observed, displaying only one instance of the data, and moving the rest to supplementary figures.

Comment #2:

It's true that we show the same datasets in multiple ways, but we believe that each panel provides unique information. For instance, in Figure 4, **a** highlights the correlations among the nodes belonging to different functional clusters, while **b** shows their spatial locations in the brain. Similar functional versus spatial information is provided in Figure 4**c** and **d**. This information is shown in more distilled forms later, in order to remain space-efficient, but their appearance in full earlier in the manuscript will be important to understanding their abbreviated presentation. For instance, Figure 4d (three panels) may be necessary for understanding what is shown in Figure 5e (one panel).

Beyond this, we feel that the methods require introduction for nonexperts in computational neuroscience, and that figures can provide a visual guide of how the data were handled. Elements of Figures 4 and 7 are dedicated to such introductions, given the broad readership of *Nature Communications*.

Of course, we are willing to be flexible in terms of the length and detail if the reviewer and editor feel that material should be moved to supplementary information.

My main concern with the graph analysis is how sensitive such an analysis is to biological noise, and therefore how meaningful changes observed are. Since the authors do not appear to have a negative control for such an effect, such as two independent populations of wt fish, it is difficult for me to tell whether the observed differences really gain more meaningful insight than the traditional correlation/clustering based methods in Figure 2 and 3.

Comment #3:

For both wt and *fmr1* mutants, datasets were acquired across numerous clutches, and analysed animals of different genotypes were siblings that were raised together. In the particular case of the wt dataset shown in Figs 1-4, each day we would run the different experimental paradigms (f20, f60, s20, s60) in a different order, attempting to collect at least one fish per stimulus train type. These are long sessions, and most often we collected one fish of each stimulus train per day. As each day involved fish from a different clutch, we believe that the similar results across days strongly suggests that we are showing a true biological phenomenon and not just noise. Furthermore, when we repeated the s20 experiments on wt animals as part of the *fmr1* experiments, we found the same clusters and distributions despite the fact the data were collected in different animals and several months later.

In the specific case of the sensitivity of the graph analysis, in response to a previous reviewer's similar concerns, we analysed our data with a leave-one-out approach to make sure that the population's responses were not being unduly driven by individual animals. This is shown for wt animals in Supplementary Figure 8e and f. Relevant to this comment and to the concerns expressed in the next question about the *fmr1* phenotype, we applied leave-one-out to both the wt and *fmr1* datasets while doing our statistical comparisons, and the significant differences were robust to this test.

We can exclude the possibility that we are simply picking up noise with our graph analysis with surrogate data, as demonstrated in Supplementary Figure 8. The fact that our data lose their graph structure when a Fourier transform is applied suggests that actual biological patterns are being detected in our main analysis.

Further evidence for the validity of the graph method comes from differences found among wildtype larvae shown different stimulus trains. The trends observed in Figures 2 and 3 are maintained and elaborated in the graph analysis found in Figure 4. Most importantly, the graph theory is sensitive to trends also seen in a bona-fide habituation process (e.g. stronger recovery at shorter ISI). From our perspective, these effects' consistency across behaviour, clustering analysis, and graph analysis lends credibility to the graph analysis both here and in the *fmr1* analysis to follow. We note similar alignments of behaviour, clustering, and graph structure in the wt vs *fmr1* analysis.

For this reason, we feel that the graph analysis is valid, but whether it provides insights beyond the clustering method is a separate issue. There are elements of the graph analysis that are not demonstrable from the clustering approach (quantifiable changes in the relationships among nodes in particular brain regions, for instance). As mentioned previously, these align with our behavioural results and also with the existing habituation literature. They provide detail, for instance, on the key role of moderately habituating neurons or the potentially pivotal role of the tectum, on how habituation may actually be playing out across this brain-wide network. Of course, these conclusions are not experimentally tested or firmly established in this manuscript, but they are only opened for discussion because we have performed graph theory, and we feel that this is sufficient value to warrant their inclusion in the paper.

Concerns with the Fmr1 mutants. From my perspective, the data shown in Figure 5a is not convincing. Based on this data alone I would not be very confident that indeed these mutants exhibit a meaningful phenotype at the behavioural level. How many different clutches of larvae went into this experiment? Has this effect been validated in different biological replicates in separate clutches? The study does report a "significant" difference in individual timepoints, but considering the large sample sizes and noise inherent in behavioural data, I am not convinced that there is a meaningful difference here based only on a p-value. This of course makes me wonder about the meaningfulness of the graph-based analyses in these mutants (though these appear to be consistent with deficits in habituation). But without a relevant negative control for these analyses (see above) it is again difficult to know how much of this difference may be due to biological noise, especially because no significant changes in the cluster distributions was found (fig 5b)

Comment #4:

We are modelling a syndrome in which the human phenotypes are subtle and variable. In this sense, we would have been surprised by and sceptical of a behavioural phenotype that is vastly more pronounced than those found in humans with FXS. We feel that a subtle phenotype, conserved with humans, contributes to the model's face validity.

Nonetheless, we need to demonstrate this subtle phenotype in a rigorous manner. We analysed these data using other statistical tests (e.g. Wilcoxon signed-rank test), and these also showed significance in the 2nd trial (we have gone with one of the more conservative corrections in the manuscript, so significance is only claimed for the 2nd trial). These data were collected across 12 clutches, and as mentioned in the response of Comment #3, this serves as a sampling method that includes clutch variability, meaning that results have to exceed normal variability to achieve significance. There are a few observations that give us faith in our interpretations. First, the significant trial does not occur in isolation. The second and third trials also show elevated responses before the response rate fades into the wt range in the fourth trial. The 11th trial also stands out, as would be expected for the decreased retention of habituation. These three trials are three out of the four (the other being the 16th) in which the *fmr1* fish approached significance. In other words, one trial (the 16th) out of 20 had an uncorrected $p < 0.05$ without a biological explanation related to habituation, matching expectations for 20 independent tests. Bonferroni correction pushed the 16th trial well above the threshold for significance, which is exactly how this correction is supposed to work. The same correction pushed the 3rd and 11th trials out of significance, while the 2nd trial (the most sensitive trial for detecting a habituation phenotype) remained strongly significant after correction. To us, these are exactly the results that you might expect for a subtle but bona fide behavioural phenotype.

We also note that in each of the trials (defining this broadly as the 2nd, 3rd, and 11th) where a habituation phenotype may be manifested, the heterozygotes have an intermediate phenotype. Because we have little statistical traction on this (showing weaker but significant significance in the 2nd trial), we do not make claims about a heterozygote phenotype or its possible mechanisms. While this effect in heterozygotes is not our main claim supporting the validity of our *fmr1* mutant phenotype, it provides one more line of evidence that the mutant phenotype is not the product of random variability.

Finally, while this phenotype is not obvious in the clustering data (which do not lend themselves well to statistical analysis), it is manifested in the graph analysis, where it occurs in just the same manner that is seen in the behavioural assay.

Overall, we feel that these diverse lines of evidence, and especially the statistically conservative analysis of the behavioural data, support the validity of the subtle phenotype that we claim.

Figure 1 - why not use color? Panel b is very confusing. Why is white and ISI not a constant number? What is the difference between white and ISI?

Comment #5:

White is the period of a white screen after the fade has finished but before the next stimulus occurs. The ISI is the entire time from one stimulus to the next, including all five phases. The ISI is slightly variable to prevent the larvae's predicting the exact onset time. Reading your comment, we see that this was not adequately explained, and we have expanded the figure legend, hopefully making this clearer. This can be found in lines 1397-1402.

Since there are colours in most of the manuscript's figures, and since these colours are maintained with particular meaning (green, blue, and red for quickly, moderately, or slowly habituating neurons, respectively; blue for wildtype and red for *fmr1*), we thought that keeping colours out of this figure made interpretation less confusing for the reader. If, taking this into account, the reviewer still prefers colours in Figure 1, we are open to adding them.

The use of the dishabituation stimulus. The authors conclude that this does not produce an effect. The potential reasons for this should be discussed. If there is no effect, then why do subsequent imaging experiments include this dishabituation stimulus?

Comment #6:

Yes, the acoustic stimulus is awkward. We had expected it to be important and interesting, but it was neither. Perhaps it wasn't loud enough. Perhaps zebrafish larvae do not dishabituate in this way. In any case, it's an unsatisfying loose end. As a side comment, when discussing our results with an expert in the habituation field at a conference, we learned that dishabituation is more difficult to elicit than we had initially realised. The reason we kept it in the subsequent imaging experiments was to try to detect dishabituation in neuronal responses regardless of

whether these were translated in a behavioural response. Unfortunately we could not convincingly detect dishabituation in our calcium imaging data either. Having learned from our initial data collection and analysis in wt animals, we reduced our stimulus train to 2 x 10 stimuli and dropped the acoustic stimulus in our analysis of the *fmr1* phenotype. As requested, we have added a brief discussion of why dishabituation has not occurred, which can be found in lines 181-186.

I was specifically asked to comment on the Author's response to previous Reviewer 4 (R4). R4's first major concern was that of a lack of technical novelty. In my opinion, the study should not be judged based on what techniques it employs, but what biological insight is gained. I am satisfied with the response of the authors to this point.

R4's second major concern is summarized by their statement

"...the results are just descriptive. There is no clear unique insight into brain function that was gained from their analyses. Their results fit with established models of information flow in the brain. The authors did not attempt to reveal underlying mechanisms, for example through some perturbation of the key regions involved."

*While this criticism is true, it could be equally applied to many neuroscience studies published in top journals (though maybe not for the zebrafish field, unfortunately for us!). The proposed manipulations would certainly be a huge boon to the study, but without a way to genetically target the relevant neurons, these studies are exceptionally challenging. The authors do in fact present a manipulation (the *fmr1* mutants), which I feel is very interesting (though I do have concerns related to the presence of a meaningful behavioural effect in these mutants, described above).*

We thank this reviewer for his/her perspectives, and for comments that allowed us to make several improvements to the manuscript.

REVIEWERS' COMMENTS

Reviewer #5 (Remarks to the Author):

I am satisfied with the rebuttal to my points, and feel like I cant offer any more substantive comments. I would like to congratulate the authors on their very nice study.

Response to reviewers:

REVIEWERS' COMMENTS

Reviewer #5 (Remarks to the Author):

I am satisfied with the rebuttal to my points, and feel like I cant offer any more substantive comments. I would like to congratulate the authors on their very nice study.

We would like to thank Reviewer #5 for the comments that helped improve our manuscript, for her/his endorsement for the manuscript acceptance and finally for the congratulations that she/he granted us for this work.